# Impact of host climate model on contrail cirrus effective radiative forcing estimates

Weiyu Zhang[1], Kwinten Van Weverberg[2,3,4], Cyril J. Morcrette[4,5], Wuhu Feng[1,6], Kalli Furtado[4,7], Paul R. Field[1,4], Chih-Chieh Chen[8], Andrew Gettelman[9], Piers M. Forster[1], Daniel R. Marsh[10], Alexandru Rap[1*]

[1]School of Earth and Environment, University of Leeds, Leeds, LS2 9JT, UK
[2]Department of Geography, Ghent University, Ghent, Belgium
[3]Royal Meteorological Institute of Belgium, Brussels, Belgium
[4]Met Office, Exeter, EX1 3PB, UK
[5]Department of Mathematics and Statistics, Exeter University, Exeter, EX4 4QE, UK
[6]National Centre for Atmospheric Science, University of Leeds, Leeds, LS2 9PH, UK
[7]Centre for Climate Research Singapore, Meteorological Service Singapore, 537054, Singapore
[8]National Center for Atmospheric Research, Boulder, CO, USA
[9]Pacific Northwest National Laboratory, Richland, WA, USA
[10]School of Physics and Astronomy, University of Leeds, Leeds, LS2 9JT, UK

*Correspondence to*: Alexandru Rap (A.Rap@leeds.ac.uk)

**Abstract.** Aviation is currently estimated to contribute ~3.5% of the net anthropogenic effective radiative forcing (ERF) of Earth's atmosphere. The largest component of this forcing comes from contrail cirrus (also with a large associated uncertainty of ~70%), estimated to be two times larger than the contribution from aviation $CO_2$ emissions. Here we implement the contrail parameterisation previously developed for the USA NCAR (National Center for Atmospheric Research) Community Atmosphere Model (CAM) in the UK Met Office Unified Model (UM). By using for the first time the same contrail parameterisation in two different host climate models, this work investigates the impact of key features of the host climate model on quantifying contrail cirrus radiative impacts. We find that differences in the background humidity (in particular ice supersaturation) in the two climate models lead to substantial differences in simulated contrail fractions, with UM values being two to three times as large as those from CAM. We also find contrasting responses in overall global cloud fraction due to air traffic, with contrails causing increases and decreases in total cloud fraction in the UM and in CAM, respectively. The different complexity of the two models' cloud microphysics schemes (i.e. single and double-moment cloud schemes in the UM and CAM, respectively) results in significant differences in the simulated changes in cloud ice water content due to aviation. When compensating the unrealistically low contrail optical depth simulated in the UM, we estimate the contrail cirrus ERF of the year 2018 to be 40.8 mW m$^{-2}$ in the UM and 60.1 mW m$^{-2}$ in CAM. While these two estimates are not entirely independent, they indicate a substantial uncertainty in contrail cirrus ERF from differences in the microphysics and radiation schemes of the two host climate models. We also find a factor of 8 uncertainty in contrail cirrus ERF due to existing uncertainty in contrail cirrus optical depth. We suggest that future work on the contrail cirrus climate impact should focus on better representing the microphysical and radiative contrail characteristics in different climate models and on improved observational constraints.

# 1 Introduction

The aviation sector has been growing rapidly over the last six decades, except for a temporary decrease in traffic caused by the measures to limit the spread of the COVID-19 pandemic (ICAO, 2023). The steady growth of civil aviation has resulted in an average increase in carbon dioxide ($CO_2$) emissions of 2% $yr^{-1}$ between 1970 and 2012, further accelerating to 5% $yr^{-1}$ from 2013 to 2018 (Lee et al., 2021). According to the International Civil Aviation Organization (ICAO) monthly monitor, passenger air traffic on most routes has reached the pre-pandemic revenue passenger kilometer (RPK) level by the end of 2023 (Icao, 2024). Furthermore, predictions based on various economic and future aircraft emission scenarios estimate that aviation fuel usage and associated $CO_2$ emissions could experience 2-3 fold increases by 2050 (Dray et al., 2022).

As recently reviewed by Lee et al. (2021), global aviation in 2018 contributed 3.5% to the total anthropogenic effective radiative forcing (ERF). While a third of the aviation ERF is estimated to be caused by $CO_2$ emissions, the majority (i.e. 2/3) is associated with non-$CO_2$ effects, including aviation induced cloudiness (contrails and contrail cirrus) and emissions of nitrogen oxides ($NO_x$), water vapour, and aerosols (in particular, soot and sulphate). Of the known aviation climate forcing contributors, contrails and contrail cirrus (i.e. spreading contrails that are no longer line-shaped) are estimated to be the largest, responsible for almost twice as much as the $CO_2$ contribution (Burkhardt and Kärcher, 2011). Contrails are line-shaped high clouds that form as a result of the mixing between the warm and moist jet engine exhaust and the cool ambient air under liquid water saturation conditions in the young plume behind the aircraft (Schumann, 1996). Schmidt (1941) and Appleman (1953) described the contrail formation process based on the thermodynamic theory. At present, water vapour primarily condenses on particles emitted by today's kerosene combusting engines. However, these emitted particles are not necessary at the contrail formation stage as particles from the ambient air could be entrained into the exhaust plume and act as condensation nuclei. When the ambient air is supersaturated with respect to ice, contrails can persist and may last for minutes up to several hours (Minnis et al., 1998). Persistent contrails may evolve into extended cirrus clouds (i.e. contrail cirrus) due to spreading and shearing (Kärcher, 2018).

Like natural cirrus, contrail cirrus changes the radiative balance of Earth in two ways. First, their ice crystals can scatter shortwave radiation back to space leading to a reduction in solar radiation reaching the Earth's surface (shortwave cooling effect). Second, contrails absorb infrared (longwave) radiation from Earth and re-emit at lower temperatures (longwave warming effect). For contrails, the longwave warming effect dominates on average, causing a net positive (warming) radiative forcing (RF) (Burkhardt and Kärcher, 2011; Rap et al., 2010a; Chen and Gettelman, 2013). Contrail cirrus can also have an impact on natural clouds as their presence changes the water budget of the surrounding atmosphere. This may partially offset the direct climate impact of contrail cirrus (Burkhardt and Kärcher, 2011) and therefore reduce the contrail cirrus climate efficacy (Bickel, 2023).

The global climate impact from aviation, including contrails and contrail cirrus, has been reviewed over the past decades. Initial assessments (e.g. Minnis et al. (1999) and Lee et al. (2009)) only considered the effect of linear contrails. Simulating and quantifying the effect of ageing and spreading contrails is challenging for models as this requires simulating the whole

contrail cirrus life cycle, including complex cloud microphysical processes, spreading, and the interaction with background cloudiness (Burkhardt and Kärcher, 2009). A best estimate for contrail cirrus ERF (including the natural cloud feedback) was only available in the latest two Intergovernmental Panel on Climate Change (IPCC) Assessment Reports (ARs), i.e., Boucher et al. (2013) and Lee et al. (2021). In the IPCC 5th AR, the ERF of contrail cirrus was estimated for 2011 as 50 mW m$^{-2}$ (5–95% likelihood range of (20, 150)) (Boucher et al., 2013). This was derived from scaling and averaging two studies, i.e., Schumann and Graf (2013) and Burkhardt and Kärcher (2011). More recently, Lee et al. (2021) provided estimates of contrail cirrus ERF of 57 mW m$^{-2}$ (5–95% likelihood range of (17, 98)) for 2018, calculated by scaling and averaging updated contrail cirrus RF and ERF estimates (Bock and Burkhardt, 2016; Burkhardt and Kärcher, 2011; Chen and Gettelman, 2013; Bickel et al., 2020; Schumann et al., 2015). Therefore, in the IPCC 6th AR, the uncertainty of the contrail cirrus ERF has been reduced compared with the IPCC 5th AR to 70% due to the development of process-based approaches simulating contrail cirrus in recent years. However, this uncertainty remains large, with the low confidence in the current best estimates of contrail cirrus ERF, reflecting the incomplete knowledge of key factors, such as the contrail spreading rate, optical depth, and radiative transfer processes (Lee et al., 2021). The 70% uncertainty results from the combined process uncertainties, simulated in a small number of available studies (Lee et al., 2021). There are three main sources of this uncertainty: (i) Upper tropospheric humidity and clouds. This largely originates from the high variability in the temporal and spatial scales of upper tropospheric ice supersaturation. (ii) The treatment of contrail cirrus and the interactions with natural clouds, and in particular the contrail ice crystal microphysical properties, contrail cirrus lifecycle and natural cloud adjustments. (iii) The radiative transfer response to contrail cirrus. This is largely due to differences in the radiation schemes across climate models, background cloud fields and their vertical overlap with contrail cirrus, homogeneity assumptions of the contrail cirrus field, and, furthermore, the presence of very small ice crystals and unknown ice crystal habits. In addition to this 70% quantified uncertainty, there is also the unquantified uncertainty due to the impact of contrails forming within natural clouds or the change in radiative transfer due to soot aerosols in contrail cirrus ice crystals.

The aviation industry has been under increasing pressure in recent years to substantially reduce its climate effect. Contrail cirrus is currently the largest aviation short-lived climate forcer and is therefore an important target for mitigation, which would make an immediate impact to the Earth's radiation budget. The decreases in the contrail occurrence and contrail cirrus ERF due to the reduction in air traffic resulting from pandemic restrictions in early 2020 have been demonstrated in several modelling and observation-based studies (Quaas et al., 2021; Digby et al., 2021; Schumann et al., 2021; Gettelman et al., 2021). A number of mitigation options have been explored to reach the aviation industry's commitment of achieving net zero $CO_2$ emissions by 2050, including technical improvements, operational management, and the use of alternative fuels (Aviation, 2020). However, the large uncertainties and poor scientific understanding in the contrail climate forcing could undermine the effectiveness of these mitigation strategies and create unintended consequences of increasing the overall climate forcing.

To date, there are only two global climate models able to simulate contrail cirrus, i.e. ECHAM (Burkhardt and Kärcher, 2009; Burkhardt and Kärcher, 2011) and the NCAR (National Center for Atmospheric Research) Community Atmosphere Model (CAM) (Chen et al., 2012; Chen and Gettelman, 2013). These models include different physical parameterisations for cloud

microphysics in general and contrail cirrus parameterisations in particular, with ECHAM simulating a separate contrail cirrus cloud class, and CAM integrating contrail ice with other ice clouds. This limits our ability to constrain the contrail cirrus ERF uncertainty.

In this study, we perform the first comparison of a contrail cirrus scheme across two global climate models (GCMs), each in its respective standard configuration. The main aim is to investigate the impact of key host climate model characteristics on contrail cirrus simulations by adapting the Chen et al. (2012) contrail cirrus CAM parameterisation for the UK Met Office Unified Model (UM) (Sellar et al., 2019). By using the same contrail parameterisation in two different host climate models, we are able to directly compare contrail cirrus estimates, therefore contributing to improving the understanding of main sources

of uncertainty in simulated contrail cirrus microphysical and optical properties, as well as the associated natural cloud responses.

    This paper is organized as follows: Section 2 provides descriptions of the UM and CAM models, the contrail parameterization, and the model setups used for the contrail simulations. Section 3 presents and analyses the simulated differences in ice supersaturation frequency, young contrail properties, cloud and radiation responses, and ERF estimates between the two

climate models. The summary and conclusions are provided in Sect. 4.

## 2 Methodology

### 2.1 The host climate models

    In this study, contrail cirrus simulations were performed with two atmospheric models: UM and CAM. Despite their different cloud microphysics and radiation schemes, both UM and CAM compare well with satellite observations in terms of simulated

cloud microphysical, macrophysical and optical properties (e.g. ice and liquid water path and specific humidity) (Jiang et al., 2012; Medeiros et al., 2023; Vignesh et al., 2020; Delanoë et al., 2011; Williams and Bodas-Salcedo, 2017).

    The UM is a numerical model of the atmosphere which is used for both weather and climate applications. The UM is coupled as the atmospheric component of past and current generations of the UK climate models, i.e., the Hadley Centre Coupled Model (HadCM), the Hadley Centre Global Environmental Model (HadGEM), and the UK Earth System Model (UKESM),

which are part of the Climate Model Intercomparison Project (CMIP) and have provided input to the IPCC ARs over the years. Cloud microphysics in the UM is parameterised by the Wilson and Ballard (1999) one-moment large-scale precipitation scheme, with extensive modifications. For instance, ice cloud parameterisations use the universal particle size distribution (PSD) of Field et al. (2007) built on a large amount of in situ measured PSDs and mass-diameter relations of Cotton et al. (2013). Cloud fraction and condensate is addressed by the PC2 large-scale cloud scheme of (Wilson et al., 2008). The impact

of convective cloudiness is represented by source terms that couple the convection scheme to PC2, based on Tiedtke (1993) and Wilson et al. (2008). In-cloud supersaturation is permitted by the model and is diagnosed by the parametrization described in Furtado and Field (2017). The parametrisation assumes that the ice cloud fraction in each gridbox is partitioned into supersaturated and sub-saturated sub-areas. The areas and RH of these regions are parameterised in terms of grid-box mean

quantities from an assumed sub-grid RH distribution. Additional complexities are introduced to handle mixed-phase and super-cool-liquid-only areas. In this scheme, there is no requirement that grid-scale RH over ice must be zero – i.e., depositional growth of ice is handled prognostically, without assuming instantaneous saturation-adjustment. Radiative transfer is calculated with the Suite of Community Radiative Transfer Codes Based on Edwards and Slingo (SOCRATES) scheme (Edwards and Slingo, 1996; Manners, 2018), using six shortwave and nine longwave radiation bands. The radiation scheme treats cloud ice crystals following Baran et al. (2016), assuming a maximum-random overlap for the vertical cloud layers.

CAM (Bogenschutz et al., 2018) is the atmospheric component of Community Earth System Model (CESM) (Danabasoglu et al., 2020). CAM version 6 (CAM6) used here employs a double-moment cloud microphysics scheme (Gettelman and Morrison, 2015; Gettelman, 2015) and has been recently updated to include rimed ice (Gettelman et al., 2019). The scheme is coupled to an aerosol microphysics and chemistry model (MAM4, (Liu et al., 2016)) and driven by a unified turbulence scheme for the boundary layer, shallow convection and large scale condensation (Bogenschutz et al., 2013; Larson, 2017). The number concentration of aerosols is connected to ice/warm cloud microphysics accounting for ice and liquid activation of cloud crystals and drops (Liu et al., 2016). In CAM6 the convective cloud scheme is based on the description of Zhang and McFarlane (1995) and Zhang et al. (1998). Ice supersaturation is allowed as described by Gettelman et al. (2010) and Gettelman et al. (2015). Saturation adjustment and condensation is performed based on the vapour pressure over liquid. Ice formation occurs only when nucleation conditions are satisfied based on the available ambient aerosols and the ice nucleation scheme of Liu and Penner (2005). Once ice is formed, a vapour deposition process occurs onto ice as described by Gettelman et al. (2010), and contrails uptake water in the same manner. The model radiation code has been updated to the Rapid Radiative Transfer Model for General Circulation Models (RRTMG) (Iacono et al., 2008). RRTMG divides the solar spectrum into 14 bands and the thermal infrared into 16 bands, with varying number of quadrature points (g points) in each of the bands. The cloud overlap is treated using the maximum-random cloud overlap assumption in RRTMG, similar to the SOCRATES radiative transfer scheme in the UM.

## 2.2 The contrail parameterisation

Previous studies have simulated the climate impact of contrails in the UM using a linear contrail scheme (Rap et al., 2010a; Rap et al., 2010b). In order to allow the UM to also simulate the spreading of linear contrails into contrail cirrus and the associated impacts on natural cirrus clouds (Burkhardt and Kärcher, 2011), here we adapt the Chen et al. (2012) CAM contrail cirrus parameterisation for the UM. This contrail parameterisation has been used in several studies on aviation climate impacts (e.g., Chen and Gettelman (2013), Gettelman et al. (2021), and Lee et al. (2021)).

In this contrail cirrus parameterisation, contrails form according to the Schmidt–Appleman criterion (Schumann, 1996) and persist in ice supersaturated regions. An empirical formula giving the critical temperature ($T_c$, in degrees Celsius) for contrail formation, as described by Schumann (1996), is

$$T_c = -46.46 + 9.43 \ln(G - 0.053) + 0.72[\ln (G - 0.053)]^2, \tag{1}$$

with $G$, in units of Pa K$^{-1}$, defined as

$$G = \frac{EI_{H_2O} \cdot c_p p}{\varepsilon Q (1-\eta)}, \tag{2}$$

where $EI_{H_2O}$ is the emission index of water vapour in $kg_{H_2O}/kg_{fuel}$, $c_p$ is the specific heat of air at constant pressure in $J\,kg_{air}^{-1}\,K^{-1}$, $p$ is the atmospheric pressure in $Pa$, $\varepsilon$ is the ratio of molecular masses of water and air in $(kg_{H_2O}mol^{-1})(kg_{air}mol^{-1})^{-1}$, $Q$ is the specific combustion heat in $J\,kg_{fuel}^{-1}$, and $\eta$ is the propulsion efficiency of the jet engine.

The critical relative humidity $RH_c$ for contrail formation depends on $G$, $T_c$ and the ambient temperature $T$ and is expressed, as in Ponater et al. (2002)

$$RH_c(T) = \frac{G \cdot (T-T_c) + e_{sat}^L(T_c)}{e_{sat}^L(T)}, \tag{3}$$

where $e_{sat}^L$ is the saturation pressure of water vapour with respect to the liquid phase.

Contrails form when the ambient temperature is below the critical temperature and the ambient relative humidity is above the critical relative humidity. Furthermore, a contrail persistence condition requires that the ambient air is supersaturated with respect to ice. In addition to the water vapour emitted by aircraft engines, ambient water vapour above ice saturation within the volume swept by aircraft is also added to the formation of contrails. This aircraft swept volume is a product of the flight path distance $d$ in m and cross-sectional area $C$ in $m^2$ (Chen et al., 2012). The contrail ice mass mixing ratio is therefore calculated as

$$M = q_t \Delta t + \frac{d \cdot C}{V} (x - x_{sat}^i), \tag{4}$$

where $q_t$ is the aviation water vapour emission mixing ratio (ratio of the mass of aircraft water vapour emission to the mass of dry air) tendency in $kg\,kg^{-1}\,s^{-1}$, $V$ is the volume of the given grid cell in $m^3$, $x$ is the ambient specific humidity (ratio of the mass of aviation water vapour emission to the total mass of air) in $kg\,kg^{-1}$, and $x_{sat}^i$ is the saturation specific humidity with respect to ice under the ambient temperature and pressure in $kg\,kg^{-1}$. The contrail fraction is calculated as the ratio of contrail ice mass mixing ratio and an empirical value for the in-cloud ice water content $ICIWC$:

$$\Delta A = \frac{M}{ICIWC/\rho_a}, \tag{5}$$

where $\rho_a$ is the density of air and $ICIWC$ is calculated as a function of temperature:

$$ICIWC(g\,m^{-3}) = e^{(6.97+0.103T(\text{°C}))} \times 10^{-3}, \tag{6}$$

The contrail parameterisation is described in detail in Chen et al. (2012), where the contrail model simulation results have been evaluated against observations.

In both host climate models (i.e. UM and CAM), contrail cirrus and natural clouds compete for water vapour available for deposition, and contrails feedback on natural cirrus cloudiness due to contrail-induced changes in moisture and temperature.

If ice supersaturation persists, the contrail cloud will take up ambient water vapour and grow. The subsequent evolution of contrails is determined by the model state, and the contrail cloud is treated no differently than any other ice cloud in the models. The contrail ice mass and fraction are added to the large-scale ice mass and cloud fraction as increments at the end of each time step.

Contrail ice number concentration is treated differently when added to natural clouds due to the different cloud microphysics

schemes in the two models. For CAM, this is a double-moment scheme (Gettelman and Morrison, 2015; Gettelman, 2015), where contrails are described by their fraction, ice water mixing ratio and ice crystal number concentration. In contrast, the UM has a one-moment cloud microphysics scheme (Wilson and Ballard, 1999), where contrails are described by their fraction and ice water mixing ratio only. This means that in the UM, the same prescribed ice number concentration is applied to both natural cirrus and contrails. The effects of the UM one-moment scheme on contrail cirrus simulations are discussed in section

3. By explicitly simulating ice crystal number concentration changes in its double-moment cloud scheme, CAM is able to overcome the contrail optical depth underestimation simulated by one-moment schemes (Kärcher et al., 2010). This underestimation remains an issue that needs to be accounted for in the UM. Another difference between the two climate models consists in the fact that while there are separate ice and snow categories in CAM, there is only one ice category in the UM (containing both ice and snow), represented by a single generic ice distribution based on a large dataset (Field et al., 2007).

In terms of air traffic inventory, we use the Aviation Environmental Design Tool (AEDT) dataset for both our UM and CAM simulations, including monthly mean distance flown and water vapour emission from air traffic for the year 2006 (Wilkerson et al., 2010). The initialised ice particles within contrails in CAM are assumed to be spherical and have a radius of 3.75 μm based on contrails aged for 20–30 min (Schröder et al., 2000; Schumann et al., 2017). In the UM, given its one-moment cloud scheme, the same PSD has to be specified for both contrail ice and natural cloud ice. The cross-sectional area $C$ of the initial

volume of contrails is assumed to be 100 m × 100 m for both CAM and UM simulations, similarly to Gettelman et al (2021), the most recent CAM contrail study. We note that using the same cross-sectional area across different spatial resolutions of the two models is expected to have only a negligible effect on young contrail properties. This is because the total contrail volume in a grid box depends not just on the cross-sectional area but also on the grid box aggregated distance flown, which ensures consistency across varying spatial resolutions.

**2.3 Simulations setup**

The models used in this study are configured in their respective standard setups, which are expected to be employed in future assessments of contrail cirrus simulations, similar to evaluations conducted for other atmospheric agents. The spatial resolution of the UM follows its CMIP6 setup (Sellar et al., 2019), while CAM6 maintains the same horizontal resolution as its CMIP6 version (Danabasoglu et al., 2020), with an adjusted vertical resolution in the Specified Dynamics (nudging) configuration to

align with MERRA-2 meteorology vertical layers. The nudging and time step settings used here reflect the default model configurations. Previous studies have quantified the impact of different configurations within a GCM, such as the impact of the microphysics scheme (Bock and Burkhardt, 2016) or the model resolution (Chen et al., 2012; Chen and Gettelman, 2013)

on contrail cirrus ERF estimates. The model configurations of the UM and CAM used in this study are described in detail below.

The CAM simulations in this study were run using CAM6 at 1.25° longitude × 0.9° latitude horizontal resolution, 56 vertical levels (~40 hPa/~1000 m in the upper troposphere and lower stratosphere (UTLS)) from the surface to about 45 km, using a model time step of 30 min. The UM simulations use the Global Atmosphere 8.0 configuration (Walters et al., 2019) at UM version 12.0, a model timestep of 20 min and a N96L85 resolution, equating to a 1.9° longitude × 2.5° latitude horizontal resolution and 85 levels (~18 hPa/~500 m in the UTLS) in the vertical with a ~85 km top.

We run 20 ensemble member simulations for each of the two models, with the imposed perturbations resulting in a slightly different atmospheric evolution for each ensemble member to represent model uncertainty. For CAM, small, random perturbations were imposed to the initial temperature fields. In the UM, ensemble parameters from different physical parameterisations were perturbed with the random-parameter method following Bowler et al. (2008) and Mccabe et al. (2016). All members in the CAM simulation are initialized on 1 January 2006 and run for one year. The UM was run for 1 year and 4
months starting from 1 September 2005 to 31 December 2006, and the first 4 months were discarded and considered to be a spin-up period. We use the Student's t-test across the ensembles to test the significance of the results.

To allow both models to capture the relatively small contrail perturbations (compared to the model internal variability in clouds and radiation) and to enhance the signal-to-noise ratio, the u and v winds field were nudged to a prescribed climatology, thus maintaining the simulated model atmosphere in a 'similar' weather state across all members and consequently reducing the
model internal variability. Winds in the UM were nudged towards the ERA5 reanalysis (Hersbach et al., 2020) on a six-hourly basis. CAM6 was run in a nudging configuration using the NASA MERRA-2 reanalysis (Gelaro et al., 2017) winds with a relaxation time scale of 24 hours, which is close to the simulation setups in Gettelman et al. (2021) to produce a similar cloud climatology as the free running CAM. We note that contrail spreading would be affected by the model wind fields, which in our UM and CAM simulations are nudged to ERA5 and MERRA2 reanalysis, respectively. Therefore, differences in the wind
fields between these two reanalyses will contribute to variations in the simulated contrail spreading across the two models.

## 3 Results

### 3.1 Ice supersaturation and contrail formation criteria in the UM and CAM6

The ice supersaturation generated by the host climate model is key for determining both the microphysical properties and lifetime of the simulated contrail cirrus. Previous evaluation studies show good agreement between simulated UM and CAM
ice supersaturation and observations (Chen et al., 2012; Irvine and Shine, 2015). The models' humidity has also been validated against observations and intercompared with other CMIP5 climate models (Jiang et al., 2012).

To evaluate the differences in the UM and CAM background meteorology, we analyse the frequency of background ice supersaturation simulated by the two models between the 100 hPa and 400 hPa pressure levels in the control simulations (i.e. without aviation contrails) based on single deterministic runs (Fig. 1a and b). The ice supersaturation frequency pattern is

similar in both models, with relatively high frequencies over mid and high latitudes below the tropopause and over the tropics in the UTLS. There is a hemispheric asymmetry in ice supersaturation frequency (larger in the Southern Hemisphere) shown in both the UM and CAM simulations. The maxima in upper tropospheric ice supersaturation in the UM expand higher up in the mid-latitudes compared to the CAM simulations, leading to the relatively higher ice supersaturation frequency in the UM compared to CAM throughout much of the UTLS. In regions with intense air traffic (coloured contour lines in Fig. 1), the

higher ice supersaturation frequency in the UM than CAM creates the potential for more persistent contrails in the UM. Both the UM and CAM capture the general pattern of ice supersaturation found in ERA5 (Fig. 1c). However, there are notable overestimations of ice supersaturation across much of the UTLS in both models compared to ERA5 which is known to have a dry bias in the UTLS (Kunz et al., 2014; Wolf et al., 2023). In high-latitude regions below the tropopause, the UM and CAM show ice supersaturation frequencies up to 50% higher than those in ERA5. In the tropical tropopause layer, CAM simulates

ice supersaturation frequencies closer to ERA5, while the UM still exhibit higher supersaturation frequencies.

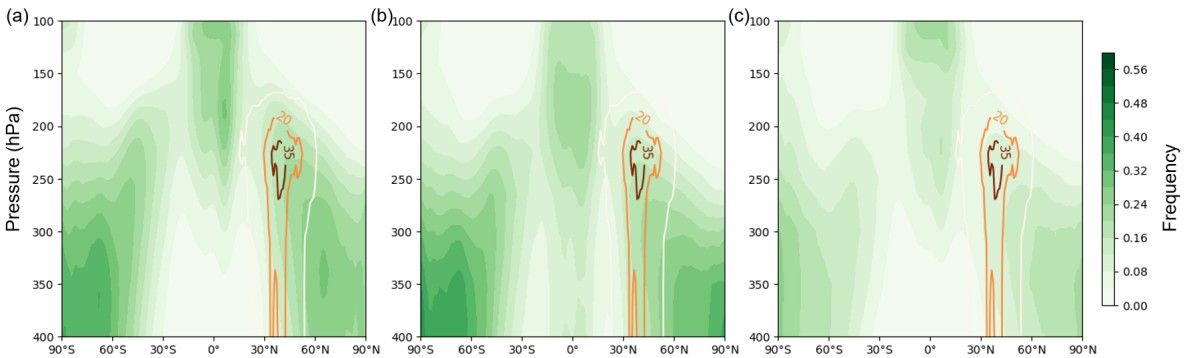

**Figure 1. Annual zonal mean frequency of background ice supersaturation simulated by the (a) UM, (b) CAM, and (c) ERA5 for the full year of 2006. The ice supersaturation frequency is calculated on a 1-hour basis. The coloured contour lines represent the annual**
**zonal means of the flight distance (in m/s) in the AEDT air traffic inventory.**

Figure 2 shows the geographical distribution of averaged background ice supersaturation between the 200 hPa and 300 hPa pressure levels (i.e. around the flight cruising levels) in the UM and CAM. The frequency of ice supersaturation in both the UM and CAM has a similar distribution pattern. There are however some important differences (illustrated in Fig. 2c), with

the UM exhibiting higher and more widespread maxima than CAM, especially in large air traffic density regions in the Northern Hemisphere, e.g. Europe and continental USA, which again can potentially facilitate persistent contrail formation. Over East Asia, which is another air traffic hotspot, the ice supersaturation frequency between the UM and CAM6 is similar. The seasonal cycle of ice supersaturation frequency difference between the UM and CAM varies spatially as shown in Fig. 2d, e, f, and g. The UM in general has higher ice supersaturation frequency except for June-July-August over Europe and

continental USA. Over East Asia, the UM has lower ice supersaturation frequency apart from March-April-May. This seasonal cycle indicates a potential seasonal cycle of the differences in contrail cirrus estimates between the two models.

The results of Schmidt-Appleman criterion satisfaction frequency averaged between the 200 hPa and 300 hPa pressure levels in the UM and CAM are shown in Fig. 3. Similar to ice supersaturation frequency, the distribution patterns between the two models are similar (Fig. 3a and b) and the UM exhibits a higher frequency of Schmidt-Appleman criterion occurrence in several regions of intense air traffic, suggesting a high propensity for contrail formation (Fig. 3c). This higher contrail formation propensity and ice supersaturation in the UM both contribute to the higher possibility for young contrail formation and persistence. There is also seasonal variation in the Schmidt-Appleman criterion satisfaction frequency difference between the two models (Fig 3a, b, c, and d). Over most part of Europe, the UM exhibits higher Schmidt-Appleman criterion satisfaction frequency except the West Europe in June-July-August and September-October-November. The UM in general has higher Schmidt-Appleman criterion satisfaction frequency mainly in December-January-February but lower in the rest seasons over continental USA.

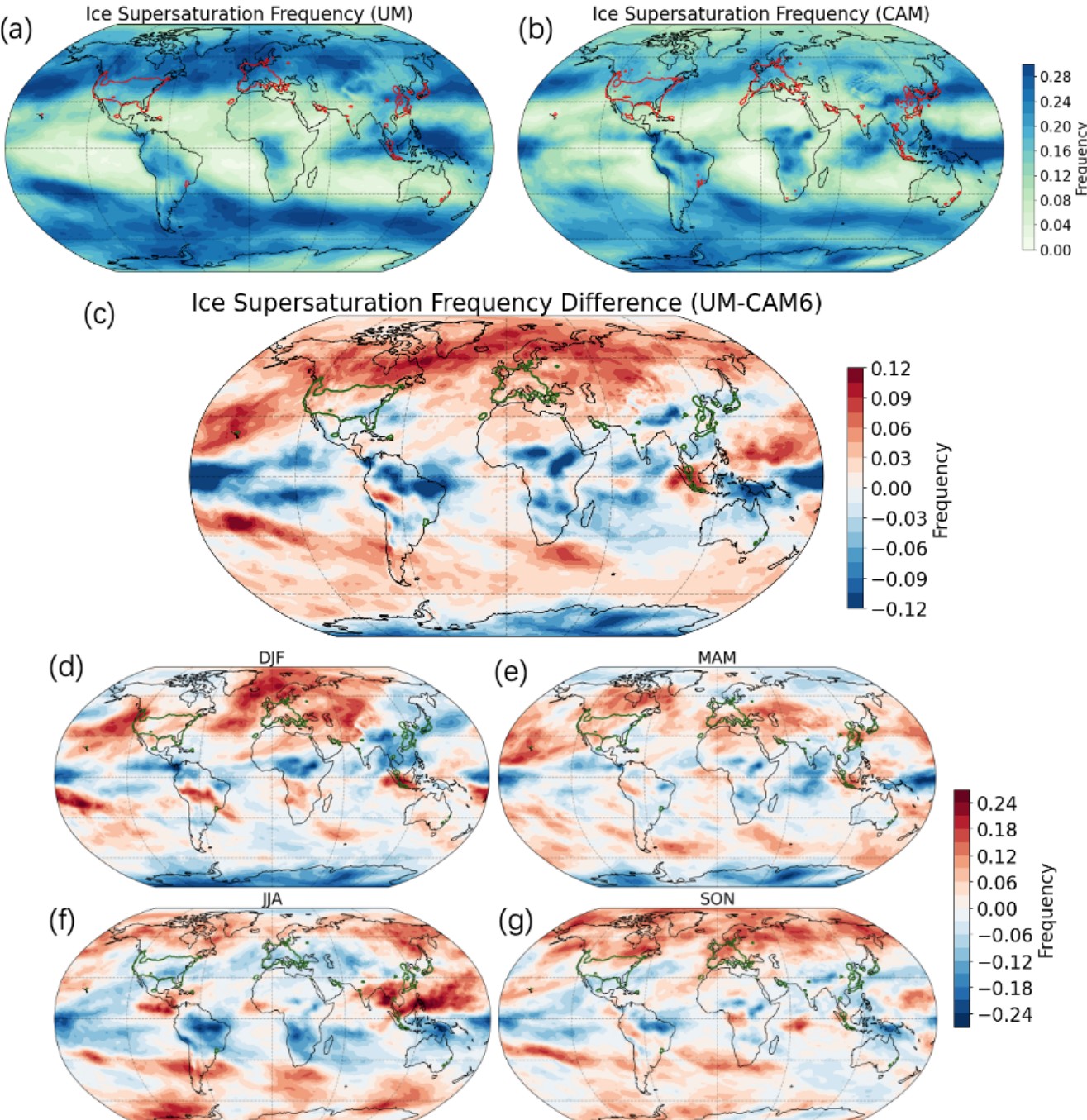

**Figure 2. Maps of the annual mean background ice supersaturation frequency averaged between the 200 hPa and 300 hPa pressure levels for 2006, generated by the (a) UM and (b) CAM. Panel (c) shows the difference in ice supersaturation frequency between the UM and CAM (UM minus CAM). Panels (d)-(g) show the seasonal mean of the ice supersaturation difference between the UM and CAM for 2006 in December–February (DJF), March–May (MAM), June-August (JJA), and September–November (SON), respectively. The green contour lines show where the mean flight distance in the AEDT air traffic inventory is over 50 meters of aggregated flight distance per second.**

The satisfaction frequencies of the Schmidt-Appleman criterion in the two models, averaged between the 200 hPa and 300 hPa pressure levels, are illustrated in Fig. 3. The overall distribution patterns between the two models are similar (Fig. 3a and b), with both showing relatively high frequencies in mid- and high-latitudes. However, the UM has a higher frequency in some regions with intense air traffic (e.g. Europe, East Asia, North Atlantic), indicating a greater likelihood of contrail formation (Fig. 3c). Combined with its generally higher ice supersaturation frequency, this increases the probability of young contrail

formation and persistence in the UM. There is also seasonal variation in the differences in Schmidt-Appleman criterion satisfaction between the two models (Fig. 3d, e, f, and g). Over East Asia and most of Europe, the UM generally shows higher frequency during all seasons, except in Western Europe during June-July-August and September-October-November. Over the continental USA, the UM generally exhibits higher frequencies in December-January-February, but lower during the rest of the year.

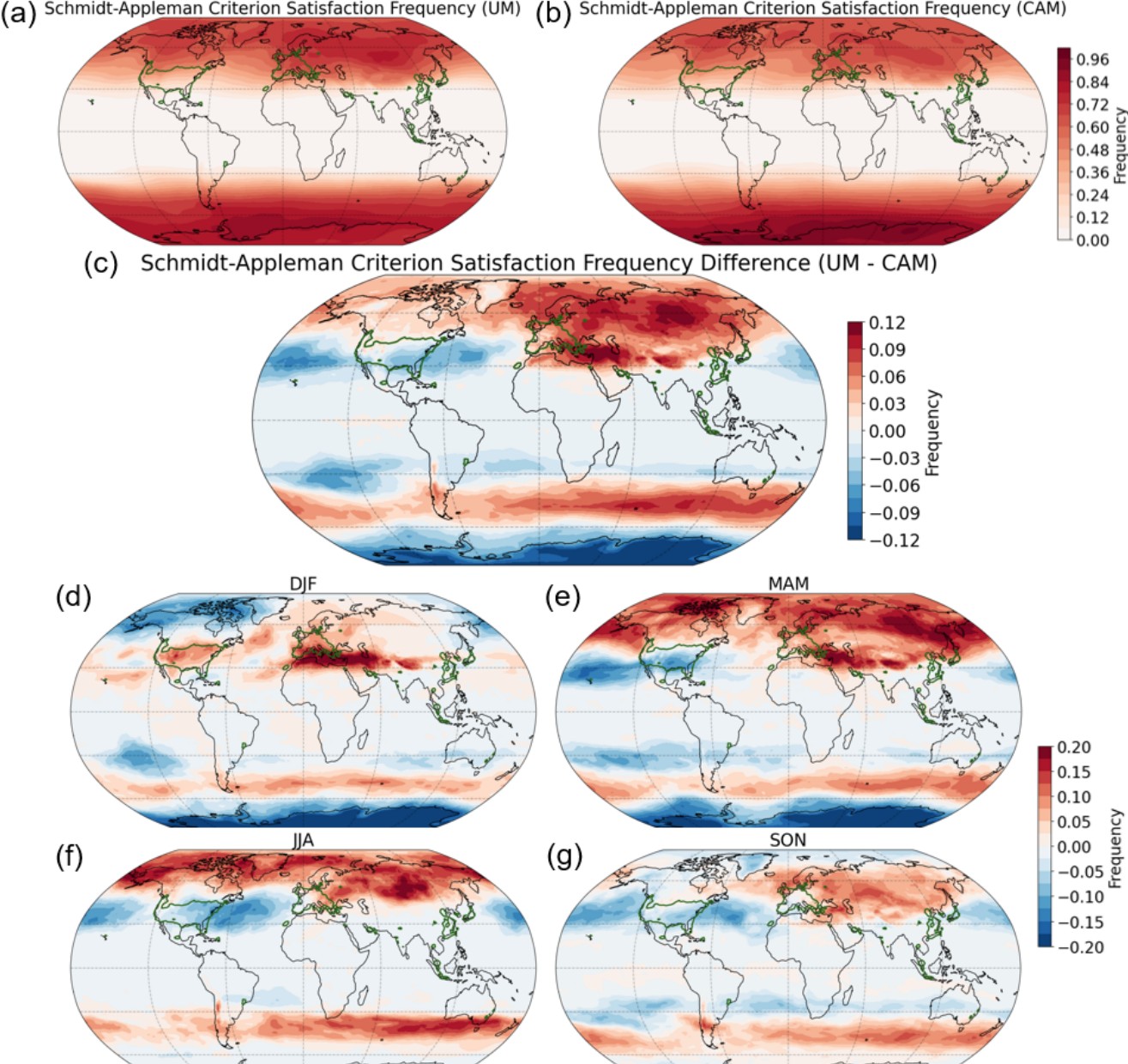

**Figure 3.** Maps of the annual mean Schmidt-Appleman criterion satisfaction frequency averaged between the 200 hPa and 300 hPa pressure levels for 2006, simulated in (a) UM and (b) CAM. Panel (c) shows the difference between the UM and CAM (UM minus CAM). Panels (d)-(g) show the seasonal mean of difference between the UM and CAM for 2006 in December–February (DJF), March–May (MAM), June-August (JJA), and September–November (SON), respectively. The green contour lines show where the mean flight distance in the AEDT air traffic inventory is larger than 50 meters of aggregated flight distance per second.



## 3.2 Simulated young contrails

The fraction and ice mass mixing ratio of young contrails, i.e. the contrail in the first model time step of its life cycle, are diagnosed by the contrail parameterisation during each model time step. Subsequently, those contrails are added at the end of the time step as increments to the natural cloud fraction and ice mass mixing ratio. The geographical distributions of the annual mean contrail cover fraction and ice water path of young contrails in the UM and CAM are shown in Fig. 4. The contrail cover fraction geographical distribution is calculated based on the random overlap assumption over vertical layers. There are many similarities in the patterns of the fraction and ice water path of young contrails simulated by the two models, since both employ the same air traffic inventory. Contrails appear mostly over the Northern Hemisphere, with maxima over the continental USA and Europe and substantial amounts over the North Atlantic corridor and East Asia. However, the magnitude of both fraction and ice water path of young contrails in the UM is substantially larger than in CAM as summarized in Table 1. We estimate a global average young contrail over fraction of 0.00012% in CAM and 0.00027% in the UM. Over the high air traffic European region defined here as 35°N - 60°N latitude and 10°W - 25°E longitude, the young contrail cover fraction averages are 0.0015% and 0.0048% in CAM and the UM, respectively. There are also large differences in the simulated young contrail ice water path. As shown in Eq. (4), the young contrail fraction and ice water mass mixing ratio are proportional to the time step length. To ensure comparability of the young contrail quantities, the CAM values were normalised to 30 minutes by a factor of 2/3 to account for the different model time step lengths (i.e. 20 minutes in the UM and 30 minutes in CAM). The normalisation was applied after the random overlap in our study, and we found that the sequence of normalisation and random overlap only had a negligible effect on the young contrail cover fraction. This normalisation may slightly underestimate CAM values, as it does not account for contrails lasting between 20 and 30 minutes period. The larger fraction and ice water path values of young contrails simulated in the UM are due to the larger frequency of ice supersaturation and Schmidt-Appleman criterion satisfaction in the UM, compared to CAM (Fig. 1, 2 and 3).

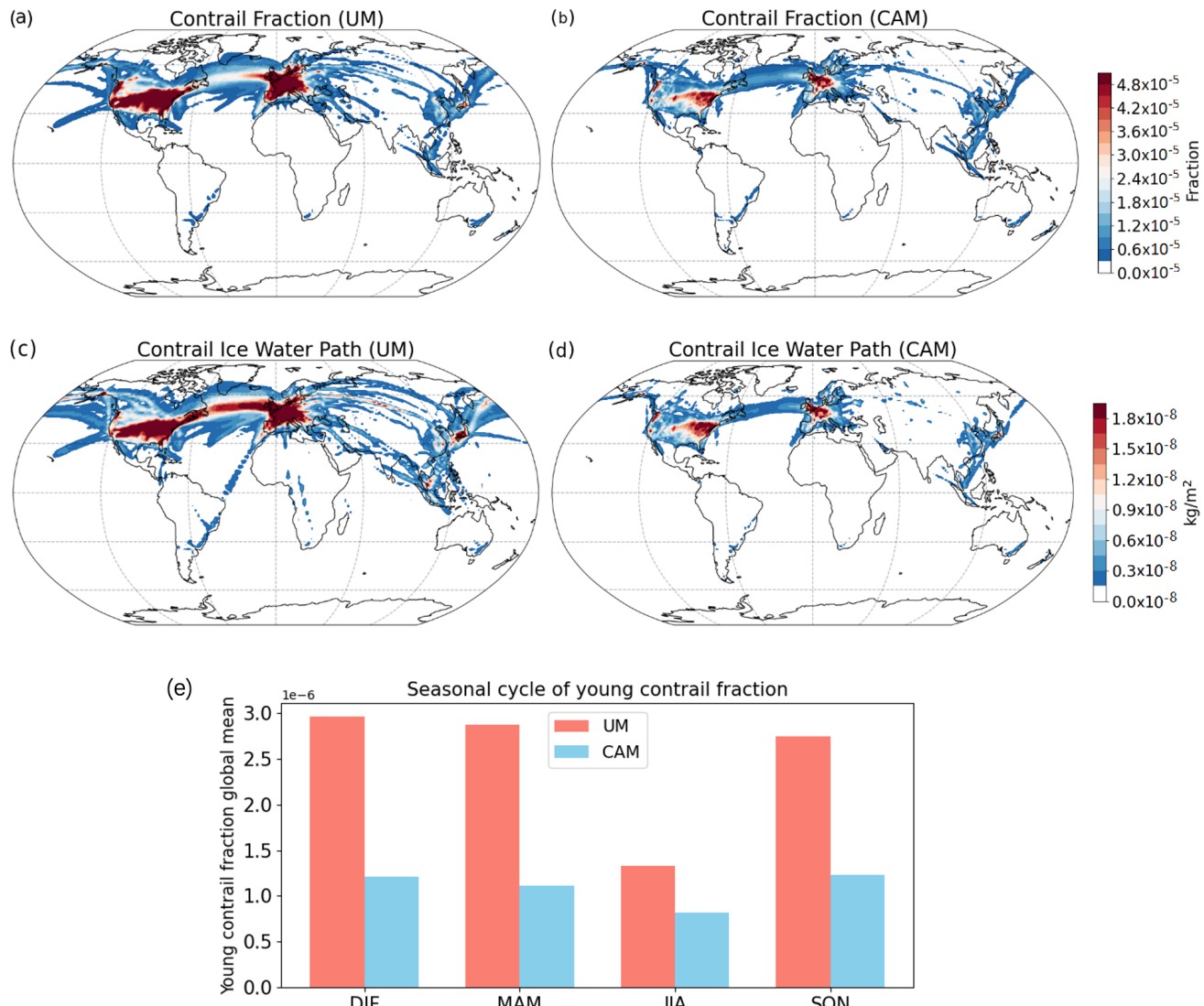

Figure 4. Annual mean young contrail cover fraction (using the random overlap assumption) from (a) UM and (b) CAM and contrail ice water path from (c) UM and (d) CAM using the AEDT air traffic inventory for 2006. Panel (e) shows the UM and CAM global seasonal mean young contrail cover fraction. The results of CAM are normalised to 20 minutes by a factor of 0.67 to account for the different model time step lengths (i.e. 20 minutes in the UM and 30 minutes in CAM).

Young contrail cover fractions of the UM and CAM have similar monthly variations, with minima and maxima in boreal summer and winter, respectively (Fig. 4e). This monthly variation is consistent with Chen et al. (2012) and other modelling and satellite observation-based studies (Bock and Burkhardt, 2016). The minima of contrail cover fraction in boreal summer primarily result from the larger upper tropospheric temperatures in the main regions of intense air traffic (i.e. Northern Hemisphere), which inhibits the formation of contrails. The young contrail cover fractions of the two models are similar to

one another during boreal summer, while during the other seasons the UM estimates are two to three times larger than those from CAM. The different magnitude of seasonal cycle in young contrail cover fraction is consistent with the seasonal cycle of ice supersaturation frequency difference mentioned in section 3.1. During the boreal summer, the global means of the young contrail cover fraction simulated by the two models are more similar, largely due to the lower ice supersaturation in the UM over some of the regions with intense air traffic (e.g., Europe and the USA).


**Table 1. Young contrail diagnostics simulated by UM and CAM using the AEDT air traffic inventory for 2006. The results of CAM are normalised to 20 minutes by a factor of 0.67 to account for the different model time step lengths (20 minutes in the UM, 30 minutes in CAM).**

| Model | Young Contrail Cover Fraction (%) | | Young Contrail Ice Water Path (kg m$^{-2}$) | |
|---|---|---|---|---|
| | Europe Mean | Global Mean | Europe Mean | Global Mean |
| UM | 0.0048 | 0.00027 | $1.79 \times 10^{-8}$ | $1.31 \times 10^{-9}$ |
| CAM | 0.0023 | 0.00012 | $0.77 \times 10^{-8}$ | $0.66 \times 10^{-9}$ |

However, our simulated young contrail cover fraction values are larger than the corresponding Chen et al. (2012) estimates, where the contrail parameterisation was implemented into the earlier CAM5 version of the model (compared to CAM6 version used in this study). Chen et al. (2012) reported a large dependence of the simulated young contrail cover fraction on the number of vertical levels in the UTLS used in CAM, with differences of up to a factor of 10 between simulations using 1000 m and 80 m vertical thickness in the UTLS. To account for this dependence, we compare the annual global mean

young contrail cover fraction of 0.00018% (unnormalized, since the time steps for both CAM5 and CAM6 are identical) from our CAM6 simulations with 56 levels overall and ~1000 m vertical level thickness in the UTLS with the corresponding value of 0.000061% estimated in CAM5 with 30 vertical levels in total but similar vertical interval in the UTLS (Chen et al., 2012). Our CAM6 simulated value is therefore ~3 times larger than the CAM5 value reported in Chen et al. (2012), indicating the effect of the different model physics and simulated ice supersaturation frequencies in the two CAM versions.

In addition, the finer UM vertical resolution in the UTLS also contributes to a larger young contrail cover fraction (Chen et al., 2012).

We note that the young contrail fraction presented here for both UM and CAM only corresponds to persistent contrails created within one model timestep and is therefore not directly comparable with the contrail fraction reported by other models. For instance, the ECHAM5 simulated contrail cirrus fraction of 0.74% for contrail cirrus with an optical depth threshold of at least

0.05 also includes contrails older than one model timestep (Bock and Burkhardt, 2016).

### 3.3 Simulated contrail impact on overall cloud fields

The UM and CAM employ distinct cloud and radiation schemes as mentioned in section 2.1. To investigate the response of the two models' cloud schemes to the presence of contrail cirrus, we analyse the averaged differences between the 20 perturbed (with contrails) and the control ensemble runs (without contrails) for each model. We also assess the statistical significance of these differences using a Student's T test, considering the change as significant when the magnitude of local perturbations exceeds two standard deviations of the 20 ensemble members (i.e. 95% confidence level).

Figure 5 shows the geographical distribution of the changes in total cloud (i.e. natural + contrail) fraction due to the presence of contrails and their feedback on natural clouds. In the UM, the presence of contrails leads to a statistically significant increase in annual global mean total cloud fraction of about 0.004%, with larger relative increases over European regions with large air traffic density, where the change amounts up to 0.4%. This overall cloud fraction response in the UM is consistent with other model studies (Bock and Burkhardt, 2016; Quaas et al., 2021). For example, the contrail cirrus simulations with the ECHAM model reported an overall relative increase in the total cloud fraction (natural cirrus + contrails) with a partly compensation by the relative decrease in natural cirrus cover (Burkhardt and Kärcher, 2011; Bickel et al., 2020). In contrast, our CAM simulations indicate a strong relative reduction in the overall cloud fraction: -0.06% globally and -0.7% over Europe. From the vertical cross section in CAM, the relative decrease in cloud fraction is mainly situated around and below the contrail formation areas (Fig. 5d). This relative reduction in cloud fraction in CAM has also been reported in Gettelman et al. (2021), where it was linked to the reduction in relative humidity caused by the local temperature increase from added contrail ice mass.

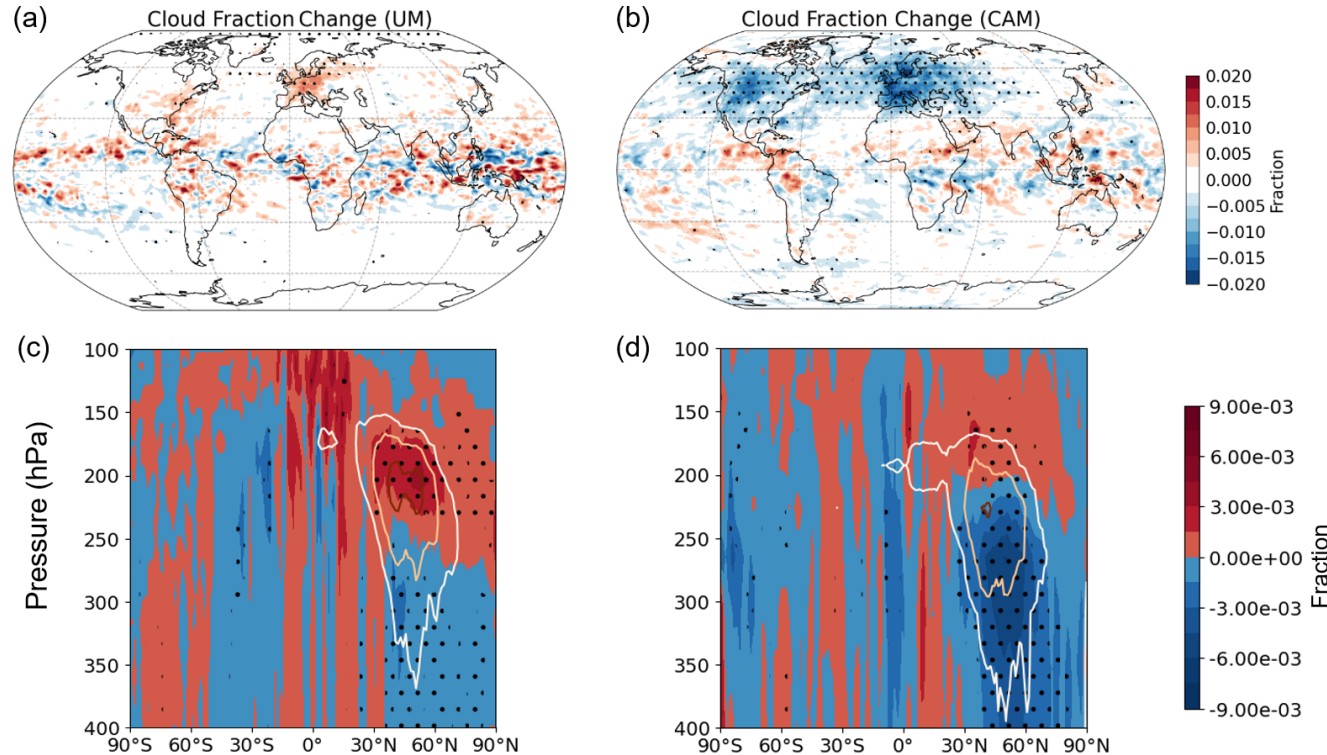

**Figure 5. Annual mean total cloud fraction changes at 220 hPa from the (a) UM and (b) CAM and annual zonal mean of cloud fraction changes caused by contrails in (c) UM and (d) CAM using the AEDT air traffic inventory for 2006. Dotted areas indicate statistically significant changes at 95% confidence level across the 20 ensemble member simulations. The contour lines represent the zonal mean young contrail fraction above 0.00002% (white), 0.0001% (yellow), and 0.0003% (green).**

The simulated impact of contrails on the overall cloud ice water content in the UM is substantially smaller, compared to the CAM simulations (Fig. 6). In terms of cloud ice water path, the simulated response in the UM (Fig. 6a) only shows some slight increases due to the presence of contrails over Europe; over other regions the changes are not significant due to small signal-to-noise ratios. There are decreases in cloud ice mass mixing ratio around the contrail formation levels, likely driven by contrail longwave heating impacts (Fig. 6c), which are overcompensated by increased cloud ice mass above and below the contrail formation levels. The small response in overall cloud ice content in the UM is very likely caused by the limitations of the one-moment cloud microphysics scheme used in the UM, where young contrails are assumed to have the same PSD as natural cirrus when added to the natural ice clouds. Also, all ice clouds, including contrails, are represented by a single ice category, with a PSD that spans the range from small crystals to large aggregates. Therefore, the contrail ice particles in the UM have much larger sizes and smaller number concentrations than those in CAM, which increases the sedimentation and sublimation rates of contrail ice particles. We note that sedimentation processes do not act as a sink term for cloud fraction in the UM, as long as some ice mass remains in the grid box. Therefore, while contrail cirrus ice mass may sediment and sublimate relatively

quickly, the contrail cirrus fraction is able to persist longer (Fig. 5a and c). In the CAM simulations, the simulated response in cloud ice water path consists in larger increases over more of the dense air traffic regions, e.g., over the European regions,

North Atlantic, and the USA (Fig. 6b). This results from increases in cloud ice mass mixing ratio both at and below contrail formation levels, likely due to the falling of contrail ice (Fig. 6d).

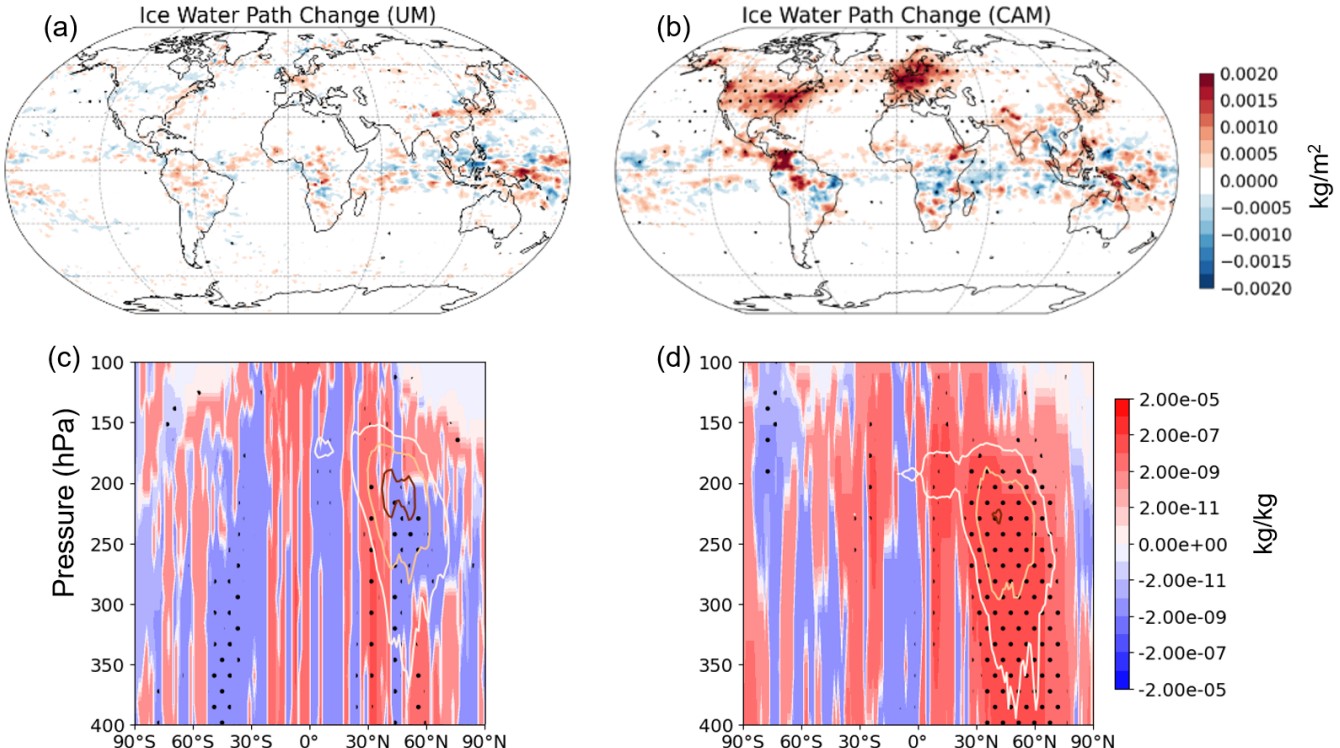

**Figure 6. Annual mean contrail-driven cloud ice water path changes in kg$^{-2}$ in the (a) UM and (b) CAM and annual zonal mean**
**contrail-driven cloud ice water mixing ratio changes in kg kg$^{-1}$ in (c) UM and (d) CAM using the AEDT air traffic inventory for 2006. Dotted areas indicate statistically significant changes at 95% confidence level across the 20 ensemble member simulations. The contour lines represent the zonal mean young contrail fraction above 0.00002% (white), 0.0001% (yellow), and 0.0003% (green).**

## 3.4 Scaling the contrail cirrus radiative response in the UM

A key limitation of this contrail cirrus scheme in the UM comes from the inability of its one-moment cloud microphysics scheme to represent, for a given contrail ice mass, realistic contrail microphysical characteristics and associated radiative effects. To address this, we adopt a method to enhance the contrail radiative response by implementing a scaling factor in the model radiation scheme for the young contrail ice mass initialized by the contrail parameterisation. The choice of this scaling factor is based on comparing the simulated UM contrail cirrus optical depth (the optical depth calculated by the changes in the

total cloud optical depth including both contrails and natural clouds caused by contrails) with other existing contrail cirrus optical depth estimates. These include both our simulation results from CAM and the published results from ECHAM (Burkhardt and Kärcher, 2011; Bock and Burkhardt, 2016). To obtain the most suitable scaling factor for each optical depth reference value, we used a trial-and-error method based on several simulations spanning scaling factors between 1000 and 20000. We used the European region (35°N – 60°N latitude and 10°W – 25°E longitude) as benchmark due to its large air traffic and therefore larger statistical significance. As a result, the scaling factors may not be representative for areas with lower air traffic density, where obtaining statistically significant results is more challenging. The range of scaling factors needed to match the range of European mean contrail cirrus optical depth reference values (i.e. 0.01-0.08) is shown in Table 2.

**Table 2. Annual global mean contrail cirrus ERFs simulated in the UM when aligning its annual European mean contrail cirrus optical depth with that of 3 different reference models. The annual mean contrail cirrus optical depths over Europe and the scaling factors used for calibrating optical depth are also included in the table.**

| Reference Model | Annual Mean Contrail Cirrus Optical Depth over Europe | Scaling factor | Global ERF of UM (mW m$^{-2}$) |
|---|---|---|---|
| CAM6 (simulated in this study) | 0.018 | 4900 | 25.7 |
| ECHAM4 (Burkhardt and Kärcher, 2011) | 0.01-0.03 | 2700-8100 | 11.1-46.0 |
| ECHAM5 (Bock and Burkhardt, 2016) | 0.05-0.08 | 13000-19000 | 76.6-91.0 |

To further compare the differences between the UM and CAM contrail cirrus simulations, in the remainder of the paper, we adopt the scaling factor (i.e. 4900) corresponding to matching the European mean contrail cirrus optical depth simulated in CAM (i.e. 0.018) – we refer to this as the scaled-UM simulation. We note that this scaling is only applied in the model's radiation scheme (where the overall cloud ice mass used to determine the cloud radiative effect therefore includes the enhanced contrail ice mass) and does not affect the actual prognostic ice mass carried through the rest of the model (e.g. dynamics or microphysics).

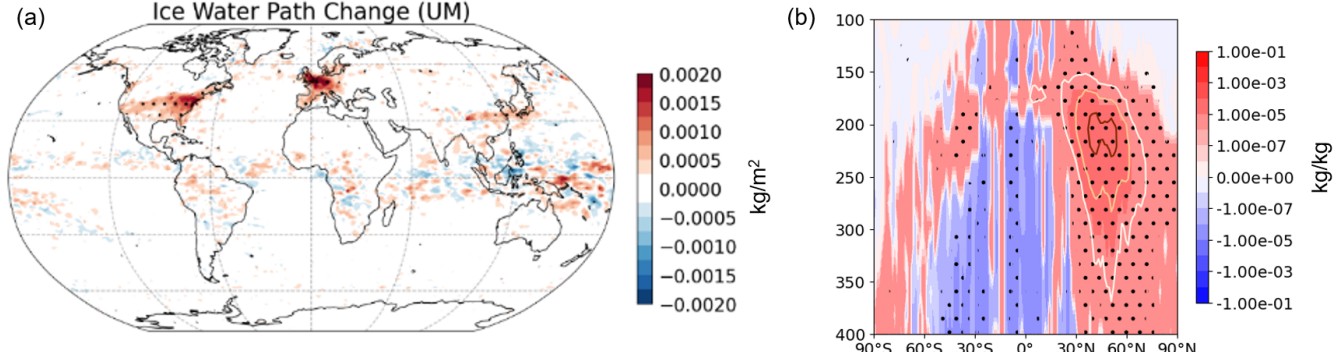

**Figure 7. (a) Annual mean simulated contrail-driven changes in cloud ice water path in kg m$^{-2}$ in the radiation scheme in scaled-UM (i.e. including the scaled-UM contrail mass) and (b) the annual zonal mean contrail-driven changes in cloud ice mass mixing ratio in kg kg$^{-1}$ in the UM radiation scheme in scaled-UM using the AEDT air traffic inventory for 2006. These values are calculated as the difference between simulations with contrails and those without contrails. Dotted areas indicate statistically significant changes at 95% confidence level. Note that the colour scale in (b) is significantly different compared to Figure 6 (c) and (d). The contour lines represent the zonal mean young contrail fraction above 0.00002% (white), 0.0001% (yellow), and 0.0003% (green).**

## 3.5 Simulated contrail cirrus effective radiative forcing

We estimate the contrail cirrus ERF, including the radiative effects of the natural cloud feedback, by contrasting simulations with and without contrails. Figure 8 shows the simulated annual mean contrail cirrus ERF in our scaled-UM and CAM simulations using the AEDT air traffic inventory for 2006. The pattern of contrail cirrus ERF in the scaled-UM is consistent with the cloud fraction (Fig. 5a) and cloud ice mass (Fig. 7a) change, with strong positive values over Europe and the USA. In CAM, the contrail cirrus ERF is consistent with the increase in cloud ice mass (Fig. 6b), but not with the decrease in cloud fraction (Fig. 5b). There is a larger increase in total cloud fraction in the cloud microphysics scheme in the scaled-UM run (not shown) compared to the unscaled-UM run (Fig. 5a). But the change in cloud ice water path in the scaled-UM run (not shown) remains similar to the unscaled-UM run (Fig. 6a). The annual global mean contrail cirrus ERF is 25.7 mW m$^{-2}$ in the scaled-UM and 52.5 mW m$^{-2}$ in the CAM simulations for 2006. In addition to the scaling of young contrails mass in the UM, the factor of 2 difference between these values is also due to the different radiative transfer schemes and different cloud microphysical process rates in the UM and CAM. By considering the range of UM scaling factors in Table 2 covering existing model estimates for European contrail cirrus optical depth, we estimate a range of UM simulated contrail cirrus ERFs between 11.1 mW m$^{-2}$ and 91.0 mW m$^{-2}$ for 2006.

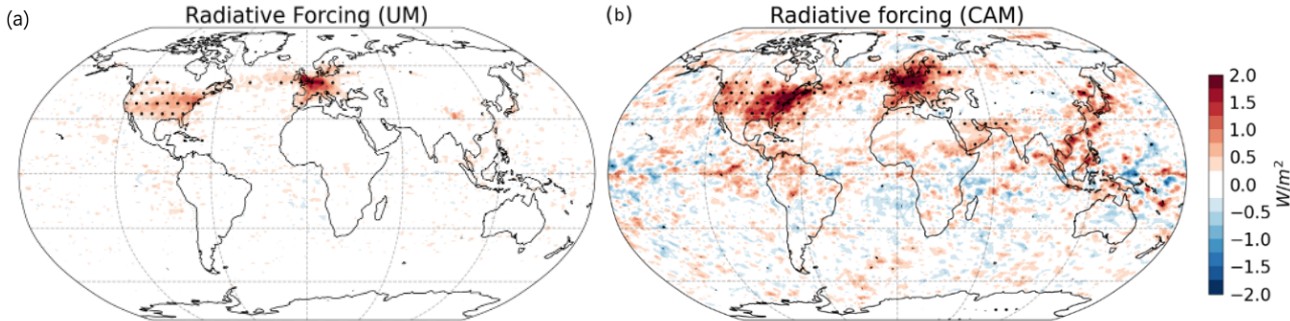

**Figure 8. Annual mean contrail cirrus ERF from (a) scaled-UM and (b) CAM using the AEDT air traffic inventory for 2006. Dotted areas indicate statistically significant changes at 95% confidence level.**

We also estimate contrail cirrus ERF for the year 2018 with both the UM and CAM to compare with the corresponding values reported in the latest IPCC AR (i.e. Lee et al. (2021)). For our 2018 simulations, we apply the 2006-2018 air traffic volume scaling factor of 1.58, based on the growth in total aircraft distance travelled reported in Lee et al. (2021), to the AEDT inventory and re-run the scaled-UM and CAM with the scaled-AEDT inventory. Our 2018 contrail cirrus ERF estimates are 40.8 mW m$^{-2}$ and 60.1 mW m$^{-2}$ from scaled-UM and CAM, respectively (Fig. 9), both within the Lee et al. (2021) 5-95% likelihood range of 17-98 mW m$^{-2}$ for contrail cirrus ERF.

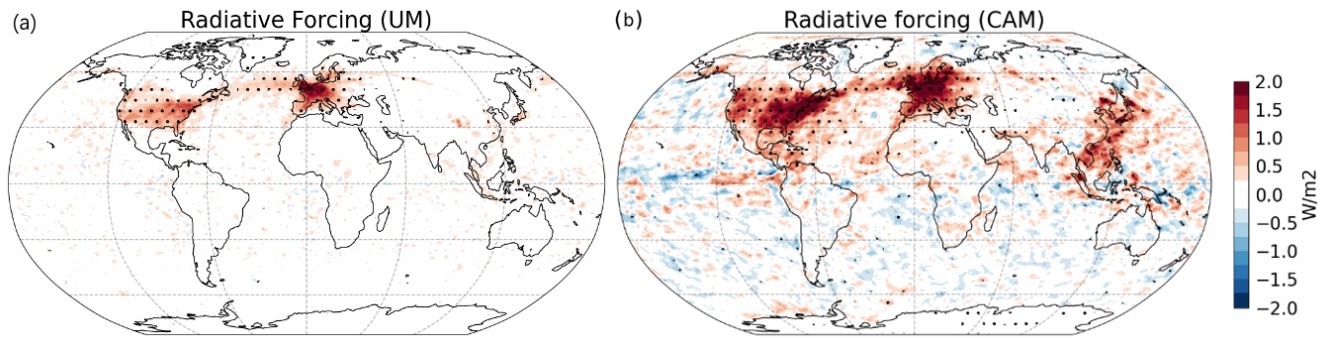

**Figure 9. Annual mean contrail cirrus ERF from (a) scaled-UM and (b) CAM using scaled AEDT air traffic inventory for 2018. Dotted areas indicate statistically significant changes at 95% confidence level.**

## 4 Summary and conclusions

In this study, we have implemented the Chen et al. (2012) contrail cirrus parameterisation from CAM in the UM. This allows to simulate for the first time contrail cirrus formation and associated cloud and radiative feedbacks in the UM. Also, by

analysing contrail cirrus simulations with the same contrail scheme in two different host climate models, we investigated the role of key model characteristics on contrail cirrus ERF uncertainty.

We found that differences in the simulated ice supersaturation frequency in the UM and CAM lead to larger young contrail ice water path and young contrail fraction in the UM compared to CAM (by a factor of 2 to 3). This highlights the critical impact of the climate models' ability to accurately represent ice supersaturation on young contrail simulations.

Our simulations indicate that the inclusion of contrails results in increased total cloud fraction (natural clouds + contrails) in the UM but decreased total cloud fraction in CAM, due to differences in the representation of microphysical processes. Also, while there is an increase in cloud ice water path in both models, this is much more pronounced in CAM. Our analysis indicates that this may be caused by the limitations of the one-moment cloud microphysics scheme and the single ice category in the UM. This results in unrealistically large sizes for contrail ice particles that affect the microphysical process rates and the life cycle of contrails in the UM. This highlights the importance of accounting for the difference in ice particle sizes between contrails and natural clouds and therefore the need for host climate models to use double-moment cloud microphysics schemes (Bock and Burkhardt, 2016).

To compensate for the incorrect representation of the contrail cirrus radiative effect in the UM, caused by the limitations from the one-moment cloud microphysics scheme, we scale up the contrail ice mass in the UM radiation scheme to match other existing European mean contrail cirrus optical depth estimates. Using this method, when matching different existing European mean contrail optical depth estimates, we obtain a range of $11.1$ mW m$^{-2}$ - $91.0$ mW m$^{-2}$ contrail cirrus ERF for 2006 in the UM. When scaling to match the Europe mean contrail cirrus optical depth simulated in CAM, the UM estimates for the contrail cirrus ERF are $25.7$ mW m$^{-2}$ for the year 2006 and $40.8$ mW m$^{-2}$ for the year 2018, while the corresponding CAM estimates are $52.5$ mW m$^{-2}$ for the year 2006 and $60.1$ mW m$^{-2}$ for the year 2018. Both of the UM and CAM contrail cirrus ERF estimates for the year 2018 are within the uncertainty range reported in Lee et al. (2021).

In conclusion, we found substantial differences between the representation of contrail cirrus formation, persistence, and radiative effects in two different climate models. Our simulations indicate: (i) a factor of 2-3 uncertainty in young contrail cover fraction due to differences in ice supersaturation frequency, (ii) contrasting (increase versus decrease) responses in total cloud fraction due to contrails, (iii) a factor of 2 uncertainty in contrail cirrus ERF due to differences in the model microphysics and radiation schemes, and (iv) a factor of 8 uncertainty in contrail cirrus ERF due to current uncertainty in simulated and/or observed regional contrail cirrus optical depths.

Another source of uncertainty arises from the differences in configurations (e.g. spatial and temporal resolutions, nudging) between the UM and CAM. In this study, both these widely used climate models are employed in their standard configurations, which are also likely to be used in future contrail studies.

Future work on contrail cirrus modelling in the UM should therefore focus on implementing (i) a contrail scheme coupled to the recently implemented double-moment cloud microphysics scheme (Field et al., 2023), but also (ii) a prognostic contrail scheme (e.g., Burkhardt and Kärcher (2009)) to allow for the detailed representation of the young contrail evolution to contrail

cirrus. Another key areas for future work consist in additional climate models to better assess the uncertainties in model physics and observation studies to better constrain the contrail cirrus radiative effects.

## Data availability

The UM code and its configuration files are subject to Crown Copyright. A licence for the UM can be requested from https://www.metoffice.gov.uk/research/approach/collaboration/unified-model/partnership. CAM6.3 used in this study is

available from https://github.com/ESCOMP/CAM/tree/cam6_3_027. Simulation output required to reproduce the main figures in this article will be available on Zenodo.

## Author contributions

WZ: Implementation of the contrail parameterisation in the UM, UM and CAM modelling, data analysis, writing (original draft preparation and editing), and conceptualisation. KVW and CJM: Implementation of the contrail parameterisation in the

UM, support with UM modelling, and writing (review and editing). WF, CC, AG, and DRM: Support with CAM modelling and writing (review and editing). KF and PRF: Support with UM modelling. PMF: writing (review and editing). AR: Implementation of the contrail parameterisation in the UM, writing (review and editing), and conceptualisation.

## Competing interests

The contact author has declared that none of the authors has any competing interests

**Acknowledgements**

Weiyu Zhang is supported by the Leeds-York-Hull Natural Environment Research Council (NERC) Doctoral Training Partnership (DTP) Panorama under grant NE/S007458/1 with the UK Met Office CASE partnership. Alexandru Rap and Piers Forster acknowledge supports from the EPSRC TOZCA grant (EP/V000772/1). Wuhu Feng was supported by the NCAS Long-Term Science programme (NE/R015244/1). We acknowledge use of the Monsoon2 system for UM simulations, a

collaborative facility supplied under the Joint Weather and Climate Research Programme, a strategic partnership between the Met Office and NERC. CAM simulations were undertaken on ARC4, part of the High Performance Computing facilities at the University of Leeds. We would like to thank Mohit Dalvi, José Rodríguez, James Manners, Harold Dyson, Andrew Clark, and last but not least John Rostron for support with and discussions on the UM.

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
