# Peer review of "Impact of host climate model on contrail cirrus effective radiative forcing estimates"

_EGUsphere, 2024_

## Referee Comment (RC1)

**Comments on "Impact of host climate model on contrail cirrus effective radiative forcing estimates" by Zhang et al. (2024)**

**General comments**

In this study by Zhang et al., the authors analyse the impact of the choice of host climate model on the simulation of contrails, their feedbacks to natural clouds and their ERF. They achieve this by running the same contrail parameterisation in two different climate models, the UK Met Office Unified Model (UM) and the NCAR Community Atmosphere Model (CAM). A main difference between the two models which is highly relevant for contrail modelling consists in the complexity of their microphysics schemes, where the UM runs a single-ice-category one-moment cloud scheme and CAM features a two-moment scheme. Overall, the authors find large dependencies of the simulated contrails and their feedbacks on the chosen host climate model with sometimes even opposite signs. In order to evaluate the response of the ERF to the choice of climate model, the simulated optical depth of young contrails in the UM is scaled up linearly to be comparable to the one simulated by CAM. This analysis also reveals a large dependency on both, the chosen host climate model and the contrail optical depth. It is concluded that the uncertainties in modelling the contrail ERF are still large.

To my knowledge, this study is the first to investigate the impact of the host climate model on contrail ERF estimates, which is an important step towards understanding and narrowing down the large uncertainty estimated by Lee et al. (2021). This fits into the scope of ACP and is of urgent scientific interest, in particular in view of plans and ongoing attempts to avoid persistent contrails. Thus, the manuscript is suitable for publication in ACP.

However, some questions, which should be addressed in the manuscript, came to my mind, while reading. Also, I have a number of comments and suggestions for improving the manuscript further and recommend that the manuscript should be revised accordingly, before it can be published.

**Specific comments**

- Line 28f: Unclear what is meant with "When accounting for the difference in cloud microphysics complexity". Suggestion: "When compensating the resulting unrealistically low contrail optical thickness in the UM"
- Line 31: The "factor of ~2" better matches the values for 2006. I suggest to either give the 2006 values in the sentence before or omit this parenthesis.
- Introduction: The authors write a lot about uncertainties in contrail cirrus RF and ERF. However, I missed a sentence on recent research suggesting a low efficacy of contrail cirrus ERF, potentially resulting in a low temperature response despite the high ERF, due to compensating slow feedbacks (e.g. Bickel 2023).
- Line 51f: This sentence seems a little out of context to me. If the intention is to state that also engines that do not emit soot can produce contrails, I suggest: "At present, the water vapour primarily condenses on particles emitted by todays kerosene combusting engines. However, these particle emissions are not necessary […]"
- Introduction: I missed the usual paragraph on the structure of the paper.
- Equations 4 & 5: If I understand this correctly, specific quantities and mass mixing ratios are mixed here. The error is certainly negligible. If this is already mentioned in Chen et al. (2012), it is probably not necessary to repeat it here. However, please be aware of this imprecision.

- Line 195: Why "100 m x 100 m"? I would assume "contrails aged for 20–30 min" to be larger in cross section. Are the choices of particle radius and contrail cross section consistent with the model time step?
- Line 204ff: What were the reasons for the different treatment of perturbations, run time and nudging between the two models? Also, I suppose nudging the temperature, but not the humidity might have strong impacts on relative humidity. Please discuss.
- Line 246: I do not see a clear signal for East Asia in JJA, especially since it is claimed in section 3.2 that East Asia may compensate the lower ice supersaturation frequency in Europe and the USA in the UM in JJA.
- Figure 2: I wonder why the difference plots show dense air traffic over the North Atlantic, while the single-model plots do not. Is this an interpolation artefact?
- Line 267: I cannot reproduce the 0.00018% for CAM from figure 3e. Is this a non-normalised value?
- Line 269ff: Was this normalisation performed before or after adding up the model levels under the random overlap assumption? In the latter case, I would (in theory) expect the reduction to be too strong.
- Line 271f: This sounds quite certain that this is the only reason. Have you checked, whether there are also differences in the frequency of contrail generation in general (Schmidt-Appleman criterion) that could also contribute to more persistent contrails?
- Lines 301ff: I do not understand the second half of this paragraph. Before, it is claimed that the young contrail fraction depends on the total number of vertical levels rather than the resolution in the UTLS. But here, young contrail fractions from two simulations with similar vertical resolution in the UTLS, but clearly different numbers of vertical levels are compared and it is proposed that the differences result from differences in the model physics rather than the different number of vertical levels.
- Line 362f: Where do the "simulated contrail ice mass" and the "contrail cirrus optical depth" come from? As far as I understood, the model is unaware of what is contrail and what is natural cirrus. Is this scaling factor only calculated and applied for "young contrails"? If this is the case, I could imagine that this contributes to the lower ERF in the UM, since the contrails would only have the enhanced optical thickness during the first model timestep of their lifecycle.
- Line 389f: Here, the ERF from the scaled UM simulations is compared with the change in cloud fraction from the unscaled simulations. I wonder, whether the cloud fraction and also the actual ice water path (not the one inside the radiation scheme) are impacted by the change in the radiation scheme.
- Line 391f: Could you comment on this apparent inconsistency? Could it be that CAM does not consider the contrail ice crystals as a cloud anymore, when they have sedimented, such that they still inhibit natural cloud formation but do not appear in the cloud fraction anymore?
- Future work: I would also conclude that more studies of this kind with other climate models and other contrail parameterisations are needed, in order to give a reliable estimate of the model uncertainty and to narrow down the uncertainty together with better observational constraints.

**Technical corrections**

As I am not a native speaker, all language corrections are rather suggestions.

- Line 30, table 2 and also later: I suggest leaving a space between "mW" and "$m^{-2}$".
- Line 38: If the authors wish to introduce the abbreviation "$CO_2$", this should be done on its first occurrence, not in line 44.

- Line 39f: "According to International […] monitor, […] has reached pre-pandemic […] level" -> "According to the International […] monitor, […] has reached the pre-pandemic […] level"
- Line 47: "(spreading […])" -> "(i.e. spreading […])"
- Line 48ff: The way the sentence is written, it appeared to me that the "cool ambient air" was "under liquid water saturation conditions". Suggestion: "Contrails are line-shaped high clouds that form whenever liquid water saturation is exceeded during the mixing between […]"
- Line 56ff and also later: Not sure, but I think "Earth" comes without an article. Please check.
- Line 99: In view of the spelling chosen throughout the manuscript, "parametrisations" is missing an "e".
- Line 123: "(Wilson et al., 2008)" should not be in parentheses.
- Line 128f: These citations also have the wrong format.
- Line 133: I think the "f" in "McFarlane" is supposed to be upper case.
- Equation 1: The bracket is meaningless, unless the square at the end is outside of the bracket (as is the case in Schumann (1996)).
- Line 148: I think the "is" is not necessary.
- Line 150ff and also throughout the manuscript: The units should not be in italics. I would also suggest to stay consistent and use either "$^{-1}$" or "/".
- Equation 3: I do not think that the bracket in the numerator improves the readability.
- Line 160: "supersaturation" -> "saturation"
- Equation 4: I suppose "$\Delta_t$" represents the time span during which the aircraft crosses the grid box (which is not explained in the paragraph below). In that case I suggest to not put "t" in subscript.
- Equation 5: "M" should be in the same font as in equation 4.
- Line 170: "ICIWC" should be in italics.
- Line 175: "condensation" -> "deposition"
- "large-scale ice and mass" -> "large-scale ice mass and cloud fraction"
- Line 181ff: Sticking with one of the terms "single-moment" or "one-moment" would probably make it easier for the reader to follow.
- Line 188: "as" is redundant.
- Line 194ff: If this is C from equation 4, I suggest inserting a "$C$" after "cross-sectional area".
- Line 200: "simulation uses" -> "simulations use"
- Line 228: "are shifted upwards" -> "extend higher up in the mid-latitudes"
- Figure 1 and also later figures: The abscissa for CAM includes the poles, while the one for the UM does not. Maybe this can be aligned.
- Line 258: Suggestion: "The contrail fraction and ice mass mixing ratio of young contrails, i.e. contrails in the first model time step of their life cycle, are diagnosed …"
- Line 261 and later: If I understand this correctly, "contrail fraction" was the fraction of a grid box filled with contrails up to this point, whereas from now on it is the fraction of a model column "covered" by contrails. If this is correct, I would suggest to introduce a new quantity "contrail coverage" or "contrail cover fraction" at this point to distinguish the two. Furthermore, I would suggest to switch the order of "fraction" and "ice water path", whenever used after "young contrail", to make clear that the young contrail ice water path is meant and not the overall ice water path.
- Line 268: The formatting of the geolocation makes it hard to read. "35° N-60° N" -> "35°N – 60°N"
- Line 281f: Chen et al. (2012) is cited twice here in the same sentence.
- Line 290: "ice supersaturation" -> "ice supersaturation frequency"
- Line 302: "vertical level height" -> "vertical layer thickness"
- Line 304: Suggestion: Add "the" before "CAM5 value"

- Line 306f: "persistent contrails within one model timestep" -> "persistent contrails created within the latest model timestep"
- Line 309: The reference to "Bock and Burkhardt (2016)" has the wrong format.
- Line 313f: This does not really make clear, how many control runs there were per model. Please rewrite, e.g. "between the 20 perturbed ensemble runs (with contrails) and the control run (without contrails) for each model."
- Line 315: "student-T test" -> "Student's t-test"
- Line 319ff: I suppose all percentages given in this paragraph are relative changes, although this is only stated once. Maybe this can be clarified also for the other percentages by adding some "relative"s, as these could also be absolute changes in cloud fraction.
- Line 327: "contrails" -> "contrail"
- Figures 4 & 5: Maybe the first two sentences of the caption can be joined with an "and", and mentioning the AEDT air traffic inventory once should then be enough. Furthermore, I missed an explanation of the contour lines in the plots. In figure 4a, "change" is the only word in the plot title that is not written with a capital.
- Line 340: "overall cloud water content" -> "cloud ice content"
- Line 344: "evaporation" -> "sublimation"
- Line 346: "evaporate" -> "sublimate"
- Line 348: The second "larger increases" seems a bit redundant.
- Line 364: The references should be in one pair of parentheses.
- Figure 6: "kg/m2" -> "kg m$^{-2}$". I would also suggest to put "(a)" and "(b)" in front of the respective panel description and not at the end. I would also be good to mention the strongly changed colour scale compared to figure 5.
- Figures 7 & 8: "forcing" in the plot title of panel b should be written with a capital. In the caption of figure 8 "(B)" should not be a capital.
- Line 418f: "young contrail fraction and ice water path" -> "young contrail ice water path and fraction" or "young contrail fraction and young contrail ice water path", as this statement is not true for the overall ice water path.
- Line 440: "models" -> "models'" or "model"
- Line 445: The format of the citation is incorrect.

**Reference:**

- Bickel, M. (2023). *Climate impact of contrail cirrus* (Doctoral dissertation, lmu).

---

## Referee Comment (RC2)

**Review of "Impact of Host Climate Model on Contrail Cirrus Effective Radiative Forcing Estimates" by Zhang et al., 2024**

In the study by Zhang et al., the influence of underlying climate models on the effective radiative forcing of contrail cirrus is examined. Two different climate models, both using the same parameterization for contrail cirrus but different microphysical schemes, are used. It turns out that differences in humidity in the UTLS region between the two models lead to significant variations in the fraction of contrail cirrus.

Quantifying the differences between the climate models in terms of their description of contrail cirrus' effective radiative forcing is crucial. Therefore, this paper aligns with the scope of ACP.

However, I have concerns about the overall structure of the comparison, which I will outline in more detail below. For this reason, I recommend a thorough revision of the manuscript before publication.

**General Comments:**

The aim of this manuscript is to address the differences in radiative forcing from contrail cirrus due to the use of different host models. The same contrail cirrus parameterization is applied to both models (CAM and UM). However, the two models differ significantly in some key parameters (as shown in the table), so it is unsurprising that the results also vary greatly. As a result, it's unclear what conclusions can be drawn from this comparison.

|  | UM | CAM |
|---|---|---|
| **Microphysics** | one-moment (Wilson and Ballard, 1999) | Two-moment (Gentleman and Morrison, 2015) |
| **Horizontal resolution** | 1.9° lon x 2.5° lat | 1.25° lon x 0.9° lat |
| **Vertical resolution** | 85 levels (500 m @ UTLS region) | 56 levels (1000 m @ UTLS region) |
| **Time step** | 20 minutes | 30 minutes |
| **Nudging** | ERA5 | NASA MERRA-2 |

This raises the question of whether the title should instead be: "Impact of the microphysical scheme on contrail cirrus effective radiative forcing estimates." Alternatively, the differences in radiative forcing could be investigated by focusing on aspects such as horizontal or vertical resolution, model time step, or even different nudging datasets. After reading the manuscript, it's unclear

to me why two climate models were used. I believe the study could have been effectively conducted using a single model with different microphysical schemes.

This leads me to question what can be learned from these results. Would it not be more useful to focus initially on a single host climate model, examining the impact of factors such as microphysics or spatial resolution? Could the spatial and temporal dimensions of the models not at least be harmonized? Doing so would make it easier to assess the differences between the two host models.

**Specific Comments:**

- **Section 2:** Since ice supersaturation is a key prerequisite for contrail cirrus formation, it would be helpful to include a more detailed description of how ice supersaturation is treated in the host climate models. For example, is ice supersaturation allowed within clouds? What about saturation adjustment?

- **L179:** You point out the important differences in microphysical schemes here, but you also discuss differences in contrail representation. This is confusing, as both models use the same contrail parameterization. Please clarify.

- **L194ff:** The cross-sectional area of the initial volume is set to 100m x 100m for both models, despite their significantly different horizontal resolutions. How does this influence the results?

- **L210ff:** Could you comment on or estimate the expected differences due to the use of different nudging datasets? Why are two different datasets used in the first place?

- **Section 3.1:** You mention good agreement between UM and CAM ice supersaturation versus observations, but Figure 1 shows clear differences in the annual zonal mean. Could you comment on this? Which model aligns more closely with observations? Over what time period is the annual zonal mean calculated—one year or more?

- **L244ff:** Why was 2006 chosen for the seasonal cycle, and how representative is that year? Which December (2005 or 2006) was used?

- **Section 3.2:** The young contrail is defined as the contrail in the first time step of its life cycle (L258). What differences can be expected if one model has a time step of 20 minutes and the other 30 minutes? Later, it's mentioned that CAM values are normalized by multiplying by 2/3. What if CAM shows no contrails after 30 minutes, but they would still be present after 20 minutes?

- **L283:** What do you mean by "high temperature in the Northern Hemisphere"? Are you referring to temperature in the UTLS region?

- **L289:** The phrase "This may be due to…" sounds speculative. Can this be substantiated?

- **L297ff:** Differences in CAM results are discussed, showing a factor of 10 when different vertical resolutions are used. How useful is it to compare CAM and UM, which have significantly different vertical resolutions?

- **L308f:** What is the temporal and spatial resolution of ECHAM5 in this case?

- **Section 3.4:** The description of the scaling factor is brief. Could you explain how you arrived at the value of 4900? Is this value representative for regions with less traffic?

- **Figure 6 Caption:** What are the "annual mean simulated contrail-driven changes" compared to?

- **L393:** The phrase "is likely due to…" sounds speculative. Can this be substantiated?

- **Summary (L416):** You mention the use of the same contrail scheme in two different host climate models. However, disentangling the differences due to one- and two-moment microphysics is challenging enough. Including different climate models with varying resolutions seems to skip a necessary step (as mentioned above).

- **L429ff and L373ff:** You write that the contrail cirrus is misrepresented in UM for understandable reasons, but it should be shown more clearly that CAM provides a more realistic representation, especially since UM's optical depth is matched to CAM's values.

- **L443 (Future Work):** It appears that microphysics is recognized as the greatest uncertainty, and improving UM with a new two-moment microphysics scheme is suggested. If microphysics is indeed the primary factor driving differences, the title of the manuscript should reflect this focus, perhaps as "Impact of Microphysics on Contrail Cirrus Radiative Forcing."

**Typos, Format, etc.:**

- **L69f:** Place the likelihood range after the unit, as in L72.
- **L77:** Avoid two parentheses directly after one another.
- **L150ff:** In LaTeX math mode, use \textrm for text and \unit{} for units. Ensure consistent typesetting and spacing between different units.
- **L123:** Omit parentheses around Wilson.
- **Figure 6:** Use "m²" for kg/m².

---

## Author Comment (AC1)

We appreciate the time and effort that Reviewer 1 has taken to review our manuscript and we thank them for their useful comments and suggestions on improving the paper. We have now addressed all the comments and made the necessary revisions to the manuscript. Please see our point-by-point responses below.

**General comments**

**In this study by Zhang et al., the authors analyse the impact of the choice of host climate model on the simulation of contrails, their feedbacks to natural clouds and their ERF. They achieve this by running the same contrail parameterisation in two different climate models, the UK Met Office Unified Model (UM) and the NCAR Community Atmosphere Model (CAM). A main difference between the two models which is highly relevant for contrail modelling consists in the complexity of their microphysics schemes, where the UM runs a single-ice-category one-moment cloud scheme and CAM features a two-moment scheme. Overall, the authors find large dependencies of the simulated contrails and their feedbacks on the chosen host climate model with sometimes even opposite signs. In order to evaluate the response of the ERF to the choice of climate model, the simulated optical depth of young contrails in the UM is scaled up linearly to be comparable to the one simulated by CAM. This analysis also reveals a large dependency on both, the chosen host climate model and the contrail optical depth. It is concluded that the uncertainties in modelling the contrail ERF are still large.**

**To my knowledge, this study is the first to investigate the impact of the host climate model on contrail ERF estimates, which is an important step towards understanding and narrowing down the large uncertainty estimated by Lee et al. (2021). This fits into the scope of ACP and is of urgent scientific interest, in particular in view of plans and ongoing attempts to avoid persistent contrails. Thus, the manuscript is suitable for publication in ACP.**

**However, some questions, which should be addressed in the manuscript, came to my mind, while reading. Also, I have a number of comments and suggestions for improving the manuscript further and recommend that the manuscript should be revised accordingly, before it can be published.**

We thank the reviewer for their positive general comments on our manuscript.

**Specific comments**

**- Line 28f: Unclear what is meant with "When accounting for the difference in cloud microphysics complexity". Suggestion: "When compensating the resulting unrealistically low contrail optical thickness in the UM"**

Thank you for pointing this out, we agree this sentence needs clarifying. We have now changed the text at lines 28-29 to:

"When compensating the unrealistically low contrail optical depth simulated in the UM, we estimate…"

**- Line 31: The "factor of ~2" better matches the values for 2006. I suggest to either give the 2006 values in the sentence before or omit this parenthesis.**

As suggested, we have now removed this parenthesis in the revised manuscript.

**- Introduction: The authors write a lot about uncertainties in contrail cirrus RF and ERF. However, I missed a sentence on recent research suggesting a low efficacy of contrail cirrus ERF, potentially resulting in a low temperature response despite the high ERF, due to compensating slow feedbacks (e.g. Bickel 2023).**

We agree that the contrail cirrus climate efficacy is extremely important. We have now added it at lines 61-64:

"Contrail cirrus can also have an impact on natural clouds as their presence changes the water budget of the surrounding atmosphere. This may partially offset the direct climate impact of contrail cirrus (Burkhardt and Kärcher, 2011) and therefore reduce the contrail cirrus climate efficacy (Bickel, 2023)."

**- Line 51f: This sentence seems a little out of context to me. If the intention is to state that also engines that do not emit soot can produce contrails, I suggest: "At present, the water vapour primarily condenses on particles emitted by todays kerosene combusting engines. However, these particle emissions are not necessary [...]"**

We agree this seemed a bit out of context and we thank the reviewer for their suggestion. We have now modified this in the manuscript at lines 51-53 as follows:

"At present, water vapour primarily condenses on particles emitted by today's kerosene combusting engines. However, these emitted particles are not necessary at the contrail formation stage as particles from the ambient air could be entrained into the exhaust plume and act as condensation nuclei."

**- Introduction: I missed the usual paragraph on the structure of the paper.**

We have now included a paper structure paragraph at lines 112-115 as follows:

"This paper is organized as follows: Section 2 provides descriptions of the UM and CAM models, the contrail parameterization, and the model setups used for the contrail simulations. Section 3 presents and analyses the simulated differences in ice supersaturation frequency, young contrail properties, cloud and radiation responses, and ERF estimates between the two climate models. The summary and conclusions are provided in Sect. 4."

**- Equations 4 & 5: If I understand this correctly, specific quantities and mass mixing ratios are mixed here. The error is certainly negligible. If this is already mentioned in Chen et al. (2012), it is probably not necessary to repeat it here. However, please be aware of this imprecision.**

Thank you for pointing this out. We agree that the distinction between specific humidity and mass mixing ratio is important, though the error is negligible in this context. We have now added the definitions of both specific humidity and mass mixing ratio in the revised manuscript for clarity at lines 183-185:
"where $q_t$ is the aviation water vapour emission mixing ratio (ratio of the mass of aircraft water vapour emission to the mass of dry air) tendency in $\mathrm{kg\,kg^{-1}\,s^{-1}}$, ..., $x$ is the ambient specific humidity (ratio of the mass of aviation water vapour emission to the total mass of air) in $\mathrm{kg\,kg^{-1}}$, ...."

**- Line 195: Why "100 m x 100 m"? I would assume "contrails aged for 20–30 min" to be larger in cross section. Are the choices of particle radius and contrail cross section consistent with the model time step?**

We acknowledge that the simulated contrails are sensitive to the choice of initial contrail parameters, as previously explored by Chen et al. (2012). In this study, the selection of the 100 m x 100 m initial

young contrail cross section and 3.75 µm initial contrail ice particle radius was made for consistency with the recent CAM6 contrail study by Gettelman et al. (2021), as noted in Sect. 2.2. Although contrails may spread to a 300 m × 300 m cross section, the 100 m × 100 m assumption refers to the area of water vapour uptake. Gettelman et al. (2021) found that assuming a larger 300 m × 300 m area was too extensive for water mass uptake, as it would imply contrails absorb all the water in that volume (A. Gettelman, personal communication). Hence, the smaller 100 m × 100 m area was used in this study. The choice of the initial particle radius is aligned with observations of contrails aged 20–30 minutes (Schröder et al., 2000; Schumann et al., 2017) and previous CAM contrail studies (Gettelman et al., 2021; Lee et al., 2021).

This has been clarified in the revised manuscript at lines 212-216:

"The initialised ice particles within contrails in CAM are assumed to be spherical and have a radius of 3.75 µm based on contrails aged for 20–30 min (Schröder et al., 2000; Schumann et al., 2017). In the UM, given its one-moment cloud scheme, the same PSD has to be specified for both contrail ice and natural cloud ice. The cross-sectional area $C$ of the initial volume of contrails is assumed to be 100 m × 100 m for both CAM and UM simulations, similarly to Gettelman et al (2021), the most recent CAM contrail study."

**- Line 204ff: What were the reasons for the different treatment of perturbations, run time and nudging between the two models?**

These choices were driven by the differences between the standard configurations of the two models, which are therefore the likely configurations to be used in future contrail simulations with the two models. Below is a summary of the reasons for these differences. We have added this discussion in Section 2.3):

- Run time: While CAM was run for 1 year, starting from 1 January 2006, since the UM standard AMIP runs start from 1 September, the UM simulation was run for 1 year and 4 months (1 September 2005 – 31 December 2006), with the first 4 months discarded as the spin-up period.
- Nudging: The nudging schemes in the UM and CAM were developed along different pathways. The use of background meteorology reanalysis differs, as the models were originally configured with different nudging reanalysis dataset. Also, the relaxation times are different , reflecting the recommended practices for each model. For the UM this consists in using the same time as the reanalysis data to maintain a stable background meteorology, while for CAM this consists in using 24 hours to allow for a similar cloud climatology as the free running CAM.

**Also, I suppose nudging the temperature, but not the humidity might have strong impacts on relative humidity. Please discuss.**

We thank the reviewer for highlighting this point as this was a mistake in the description of the methodology in the original manuscript. Neither temperature or humidity is actually nudged in the runs, therefore allowing the hydrologic cycle, including contrail and ice cloud formation processes, to operate freely. This has now been corrected in the revised version at lines 242-243 as follows:

"To allow both models to capture the relatively small contrail perturbations (compared to the model internal variability in clouds and radiation) and to enhance the signal-to-noise ratio, the u and v wind fields were nudged to a prescribed climatology, …"

As for the effect of nudging temperature on humidity, this is expected to be minimal as demonstrated in Sect. 3.3 of Gettelman et al. (2021).

**- Line 246: I do not see a clear signal for East Asia in JJA, especially since it is claimed in section 3.2 that East Asia may compensate the lower ice supersaturation frequency in Europe and the USA in the UM in JJA.**

Thank you for pointing this out. We agree the explanation provided was probably misleading. We have now changed this in the revised manuscript at lines 357-359 to clarify as follows:

"During the boreal summer, the global means of the young contrail cover fraction simulated by the two models are more similar, largely due to the lower ice supersaturation in the UM over some of the regions with intense air traffic (e.g., Europe and the USA)."

**- Figure 2: I wonder why the difference plots show dense air traffic over the North Atlantic, while the single-model plots do not. Is this an interpolation artefact?**

Thank you for pointing this out, it was due to an error in the code used to draw the contour lines of air traffic density in the plot. We have now corrected this in Figure 2 in the revised version.

**- Line 267: I cannot reproduce the 0.00018% for CAM from figure 3e. Is this a non-normalised value?**

Thank you. We are sorry for the confusion – this mistake has now been corrected in the revised manuscript. The correct value of young contrail fraction global mean is  0.00012% after normalisation. This has been updated in Table 1 in the revised manuscript.

In addition, to further clarify and avoid confusions, we also now mention the unnormalized value of 0.00018% at Lines 369-373:

"To account for this dependence, we compare the annual global mean young contrail cover fraction of 0.00018% (unnormalized, since the time steps for both CAM5 and CAM6 are identical) from our CAM6 simulations with 56 levels overall and ~1000 m vertical level thickness in the UTLS with the corresponding value of 0.000061% estimated in CAM5 with 30 vertical levels in total but similar vertical interval in the UTLS (Chen et al., 2012)."

**- Line 269ff: Was this normalisation performed before or after adding up the model levels under the random overlap assumption? In the latter case, I would (in theory) expect the reduction to be too strong.**

The normalisation was applied after the random overlap calculation. We compared the results of applying the normalization factor both before and after the random overlap calculation, and only found a negligible difference (i.e. 0.0001210% compared to 0.0001208%). This has now been clarified at lines 337-339 as follows:

"The normalisation was applied after the random overlap in our study, and we found that the sequence of normalisation and random overlap only had a negligible effect on the young contrail cover fraction."

**- Line 271f: This sounds quite certain that this is the only reason. Have you checked, whether there are also differences in the frequency of contrail generation in general (SchmidtAppleman criterion) that could also contribute to more persistent contrails?**

Thank you for this suggestion. We have now performed additional analysis to investigate this further by comparing the frequency of satisfying the Schmidt-Appleman criterion in the two models. This has now been included in the manuscript as Fig. 3 and at lines 305-314.

"The satisfaction frequencies of the Schmidt-Appleman criterion in the two models, averaged between the 200 hPa and 300 hPa pressure levels, are illustrated in Fig. 3. The overall distribution patterns between the two models are similar (Fig. 3a and b), with both showing relatively high frequencies in mid- and high-latitudes. However, the UM has a higher frequency in some regions with intense air traffic (e.g. Europe, East Asia, North Atlantic), indicating a greater likelihood of contrail formation (Fig. 3c). Combined with its generally higher ice supersaturation frequency, this increases the probability of young contrail formation and persistence in the UM. There is also seasonal variation in the differences in Schmidt-Appleman criterion satisfaction between the two models (Fig. 3d, e, f, and g). Over East Asia and most of Europe, the UM generally shows higher frequency during all seasons, except in Western Europe during June-July-August and September-October-November. Over the continental USA, the UM generally exhibits higher frequencies in December-January-February, but lower during the rest of the year."

[Figure]

**Figure 3. Maps of the annual mean Schmidt-Appleman criterion satisfaction frequency averaged between the 200 hPa and 300 hPa pressure levels for 2006, simulated in (a) UM and (b) CAM. Panel (c) shows the difference between the UM and CAM (UM minus CAM). Panels (d)-(g) show the seasonal mean of difference between the UM and CAM for 2006 in December–February (DJF), March–May (MAM), June-August (JJA), and September–November (SON), respectively. The green contour lines show where the mean flight distance in the AEDT air traffic inventory is larger than 50 meters of aggregated flight distance per second.**

**- Lines 301ff: I do not understand the second half of this paragraph. Before, it is claimed that the young contrail fraction depends on the total number of vertical levels rather than the resolution in the UTLS. But here, young contrail fractions from two simulations with similar vertical resolution in the UTLS, but clearly different numbers of vertical levels are compared and it is proposed that the differences result from differences in the model physics rather than the different number of vertical levels.**

Sorry for the confusion, we have now changed the text to clarify what we mean here. The vertical resolution in the UTLS is the critical factor, as this is where contrails mostly form. In the paragraph in question, we compare results from CAM5 and CAM6 simulations, with different total numbers of vertical levels but similar vertical resolution in the UTLS. So the differences shown are more likely due to variations in the model physics, rather than the number of vertical levels. This has been revised in the new version at lines 367-374 as follows:

"Chen et al. (2012) reported a large dependence of the simulated young contrail cover fraction on the number of vertical levels in the UTLS used in CAM, with differences of up to a factor of 10 between simulations using 1000 m and 80 m vertical thickness in the UTLS. To account for this dependence, we compare the annual global mean young contrail cover fraction of 0.00018% (unnormalized, since the time steps for both CAM5 and CAM6 are identical) from our CAM6 simulations with 56 levels overall and ~1000 m vertical level thickness in the UTLS with the corresponding value of 0.000061% estimated in CAM5 with 30 vertical levels in total but similar vertical interval in the UTLS (Chen et al., 2012). Our CAM6 simulated value is therefore ~3 times larger than the CAM5 value reported in Chen et al. (2012), indicating the effect of the different model physics and simulated ice supersaturation frequencies in the two CAM versions."

**- Line 362f: Where do the "simulated contrail ice mass" and the "contrail cirrus optical depth" come from? As far as I understood, the model is unaware of what is contrail and what is natural cirrus. Is this scaling factor only calculated and applied for "young contrails"? If this is the case, I could imagine that this contributes to the lower ERF in the UM, since the contrails would only have the enhanced optical thickness during the first model timestep of their lifecycle.**

Yes, that is correct. The 'simulated contrail ice mass' refers to the young contrail ice mass initialized by the contrail parameterization, and the 'contrail cirrus optical depth' refers to the optical depth calculated by the changes in the total cloud optical depth (including both contrails and natural clouds) caused by contrails. So the scaling of the young contrail ice mass does indeed contribute to the smaller contrail cirrus ERF, as it only affects the early stages (i.e. first timestep) of the contrail lifecycle.

We have now clarified 'simulated contrail ice mass' and 'contrail cirrus optical depth' in the revised manuscript at lines 432-436:

"…, we adopt a method to enhance the contrail radiative response by implementing a scaling factor in the model radiation scheme for the young contrail ice mass initialized by the contrail parameterisation. The choice of this scaling factor is based on comparing the simulated UM contrail cirrus optical depth (the optical depth calculated by the changes in the total cloud optical depth including both contrails and natural clouds caused by contrails) with other existing contrail cirrus optical depth estimates."

The effect on UM ERF is now clarified at lines 473-475:

"In addition to the scaling of young contrails mass in the UM, the factor of 2 difference between these values can be attributed to the different radiative transfer schemes and different cloud microphysical process rates in the UM and CAM."

**- Line 389f: Here, the ERF from the scaled UM simulations is compared with the change in cloud fraction from the unscaled simulations. I wonder, whether the cloud fraction and also the actual ice water path (not the one inside the radiation scheme) are impacted by the change in the radiation scheme.**

Our intent here is to highlight the consistency between the cloud fraction patterns in the unscaled run and the contrail cirrus ERF in the scaled run. But the reviewer is correct - there is a larger increase in total cloud fraction in the cloud microphysics scheme in the scaled run (Fig. 1a below) compared to the unscaled run (Fig. 5a in the manuscript). However, the change in cloud ice water path in the scaled run (Fig. 1b below) remains similar to the unscaled run (Fig. 6a in the manuscript).

[Figure]

Figure 1. Annual mean (a) total cloud fraction changes at 220 hPa and (b) cloud ice water path changes in cloud microphysics from the scaled-UM

This has now been clarified at lines 470-472 as follows:

"There is a larger increase in total cloud fraction in the cloud microphysics scheme in the scaled-UM run (not shown) compared to the unscaled-UM run (Fig. 5a). But the change in cloud ice water path in the scaled-UM run (not shown) remains similar to the unscaled-UM run (Fig. 6a)."

**- Line 391f: Could you comment on this apparent inconsistency? Could it be that CAM does not consider the contrail ice crystals as a cloud anymore, when they have sedimented, such that they still inhibit natural cloud formation but do not appear in the cloud fraction anymore?**

The reason for this is explained in Sect 3.3 of Gettelman et al. (2021). We also mention this in our manuscript at lines 397-398:

"This relative reduction in cloud fraction in CAM has also been reported in Gettelman et al. (2021), where it was linked to the reduction in relative humidity caused by the local temperature increase from added contrail ice mass."

**- Future work: I would also conclude that more studies of this kind with other climate models and other contrail parameterisations are needed, in order to give a reliable estimate of the model uncertainty and to narrow down the uncertainty together with better observational constraints.**

Thank you for this point. We have now added this in the revised manuscript at lines 530-531:

"Another key areas for future work consist in additional climate models to better assess the uncertainties in model physics and observation studies to better constrain the contrail cirrus radiative effects."

**Technical corrections**

Thank you for the detailed comments on the technical corrections. They have now all been addressed in the revised manuscript.

**Reference**

Chen, C. C., Gettelman, A., Craig, C., Minnis, P., and Duda, D. P.: Global contrail coverage simulated by CAM5 with the inventory of 2006 global aircraft emissions, J. Adv. Model. Earth Syst., 4, https://doi.org/10.1029/2011MS000105, 2012.

Gettelman, A., Chen, C. C., and Bardeen, C. G.: The climate impact of COVID-19-induced contrail changes, Atmos. Chem. Phys., 21, 9405-9416, 10.5194/acp-21-9405-2021, 2021.

Lee, D. S., Fahey, D. W., Skowron, A., Allen, M. R., Burkhardt, U., Chen, Q., Doherty, S. J., Freeman, S., Forster, P. M., Fuglestvedt, J., Gettelman, A., De León, R. R., Lim, L. L., Lund, M. T., Millar, R. J., Owen, B., Penner, J. E., Pitari, G., Prather, M. J., Sausen, R., and Wilcox, L. J.: The contribution of global aviation to anthropogenic climate forcing for 2000 to 2018, Atmospheric Environment, 244, 117834, https://doi.org/10.1016/j.atmosenv.2020.117834, 2021.

Schröder, F., Kärcher, B., Duroure, C., Ström, J., Petzold, A., Gayet, J.-F., Strauss, B., Wendling, P., and Borrmann, S.: On the Transition of Contrails into Cirrus Clouds, J. Atmos. Sci., 57, 464-480, https://doi.org/10.1175/1520-0469(2000)057<0464:OTTOCI>2.0.CO;2, 2000.

Schumann, U., Baumann, R., Baumgardner, D., Bedka, S. T., Duda, D. P., Freudenthaler, V., Gayet, J. F., Heymsfield, A. J., Minnis, P., Quante, M., Raschke, E., Schlager, H., Vázquez-Navarro, M., Voigt, C., and Wang, Z.: Properties of individual contrails: a compilation of observations and some comparisons, Atmos. Chem. Phys., 17, 403-438, 10.5194/acp-17-403-2017, 2017.

---

## Author Comment (AC2)

We appreciate the time and effort that Reviewer 2 has taken to review the manuscript and we thank them for their useful comments and suggestions on how to improve the paper. We believe we have now addressed their comments and made the revisions described in our point-by-point responses below.

**General Comments:**

**The aim of this manuscript is to address the differences in radiative forcing from contrail cirrus due to the use of different host models. The same contrail cirrus parameterization is applied to both models (CAM and UM).**

**However, the two models differ significantly in some key parameters (as shown in the table), so it is unsurprising that the results also vary greatly. As a result, it's unclear what conclusions can be drawn from this comparison.**

**This raises the question of whether the title should instead be: "Impact of the microphysical scheme on contrail cirrus effective radiative forcing estimates." Alternatively, the differences in radiative forcing could be investigated by focusing on aspects such as horizontal or vertical resolution, model time step, or even different nudging datasets. After reading the manuscript, it's unclear to me why two climate models were used. I believe the study could have been effectively conducted using a single model with different microphysical schemes.**

**This leads me to question what can be learned from these results. Would it not be more useful to focus initially on a single host climate model, examining the impact of factors such as microphysics or spatial resolution? Could the spatial and temporal dimensions of the models not at least be harmonized? Doing so would make it easier to assess the differences between the two host models.**

We believe our study brings two key contributions:

- The first contrail cirrus scheme in the UK Met Office Unified Model (UM): This work marks the first implementation of a contrail cirrus parameterisation in the UM, making the UM the third global climate model (GCM) capable of simulating the climate impacts of contrail cirrus. This addition helps to address a current limitation in contrail cirrus effective rafiative forving (ERF) assessments, which are presently constrained by the small number of GCMs—currently only ECHAM and Community Atmosphere Model (CAM). While the UM is not yet able to provide a fully independent contrail cirrus ERF estimate, it has the potential to achieve this with the future adoption of a two-moment cloud scheme.
- A comparison of contrail cirrus simulations with two widely used climate models: Despite various sources of uncertainty in contrail cirrus ERF estimates, the uncertainty stemming from the use of different host GCMs has been highlighted in the latest IPCC report. Our study reveals significant differences in contrail simulations between different host GCMs, offering new insights into this area of uncertainty.

This has now been clarified in lines 105-111 of the revised manuscript:

"In this study, we perform the first comparison of a contrail cirrus scheme across two global climate models (GCMs), each in its respective standard configuration. The main aim is to investigate the impact of key host climate model characteristics on contrail cirrus simulations by adapting the Chen et al. (2012) contrail cirrus CAM parameterisation for the UK Met Office Unified Model (UM) (Sellar et al., 2019). By using the same contrail parameterisation in two different host climate models, we are able to directly compare contrail cirrus estimates, therefore contributing to improving the

understanding of main sources of uncertainty in simulated contrail cirrus microphysical and optical properties, as well as the associated natural cloud responses."

However, we do not attempt to provide a comprehensive analysis of the sources of uncertainty in GCM contrail cirrus ERF estimates. Previous work has investigated the role of the GCM's microphysics scheme (Bock and Burkhardt, 2016) or the model resolution (Chen and Gettelman, 2013; Chen et al., 2012) on contrail cirrus ERF. Our study provides the first comparison of the contrail cirrus scheme in two GCMs, each in their respective standard configurations. Similarly to investigations of the impact of other atmospheric agents (e.g. aerosols (Ratcliffe et al., 2024; Henry et al., 2023) and ozone (Brown et al., 2024; Son et al., 2018; Brown-Steiner et al., 2015)), these standard configurations are likely to be used in future assessments of contrail cirrus ERF. For spatial resolution, the UM configuration is consistent with its CMIP6 setup (Sellar et al., 2019), while CAM6 maintains the same horizontal resolution as its CMIP6 version (Danabasoglu et al., 2020) with an adjusted vertical resolution in CAM6 Specified Dynamic compset (nudging) to be aligned with the MERRA2 meteorology vertical layers. The nudging and time step settings are also the models' default configurations.

While we do not aim to provide a comprehensive analysis of the role of different configurations, in the revised version of the manuscript we further clarify the discussion of these effects at lines 221-229 as follows:

"The models used in this study are configured in their respective standard setups, which are expected to be employed in future assessments of contrail cirrus simulations, similar to evaluations conducted for other atmospheric agents. The spatial resolution of the UM follows its CMIP6 setup (Sellar et al., 2019), while CAM6 maintains the same horizontal resolution as its CMIP6 version (Danabasoglu et al., 2020), with an adjusted vertical resolution in the Specified Dynamics (nudging) configuration to align with MERRA-2 meteorology vertical layers. The nudging and time step settings used here reflect the default model configurations. Previous studies have quantified the impact of different configurations within a GCM, such as the impact of the microphysics scheme (Bock and Burkhardt, 2016) or the model resolution (Chen et al., 2012; Chen and Gettelman, 2013) on contrail cirrus ERF estimates. The model configurations of the UM and CAM used in this study are described in detail below."

**Specific Comments:**

• **Section 2: Since ice supersaturation is a key prerequisite for contrail cirrus formation, it would be helpful to include a more detailed description of how ice supersaturation is treated in the host climate models. For example, is ice supersaturation allowed within clouds? What about saturation adjustment?**

Thank you for this point. We now add a more detailed description of ice supersaturation treatment within the models at lines 131-136 in the revised manuscript:

"In-cloud supersaturation is permitted by the model and is diagnosed by the parametrization described in Furtado and Field (2017). The parametrisation assumes that the ice cloud fraction in each gridbox is partitioned into supersaturated and sub-saturated sub-areas. The areas and RH of these regions are parameterised in terms of grid-box mean quantities from an assumed sub-grid RH distribution. Additional complexities are introduced to handle mixed-phase and super-cool-liquid-only areas. In this scheme, there is no requirement that grid-scale RH over ice must be zero — i.e., depositional growth of ice is handled prognostically, without assuming instantaneous saturation-adjustment."

At lines 147-151:

"Ice supersaturation is allowed as described by Gettelman et al. (2010) and Gettelman et al. (2015). Saturation adjustment and condensation is performed based on the vapour pressure over liquid. Ice formation occurs only when nucleation conditions are satisfied based on the available ambient aerosols and the ice nucleation scheme of Liu et al (2005). Once ice is formed, a vapour deposition process occurs onto ice as described by Gettelman et al. (2010), and contrails uptake water in the same manner."

**• L179: You point out the important differences in microphysical schemes here, but you also discuss differences in contrail representation. This is confusing, as both models use the same contrail parameterization. Please clarify.**

Sorry for the confusion and thank you for pointing this out. While both models use the same contrail parameterisation, the difference lies in how contrail ice number concentration is handled when added to natural clouds. In the UM, which uses a one-moment cloud microphysics scheme, the contrail ice number concentration is not explicitly specified when added to natural clouds.

We have now rephrased the sentence for clarity at lines 199-200 as follows:

"Contrail ice number concentration is treated differently when added to natural clouds due to the different cloud microphysics schemes in the two models."

**• L194ff: The cross-sectional area of the initial volume is set to 100m x 100m for both models, despite their significantly different horizontal resolutions. How does this influence the results?**

The difference in horizontal resolution does not have a big impact on the young contrail results, as long as the background meteorology conditions stay similar. This can be explained as follows:

Here is the Eq (4) in the manuscript for the parameterisation of contrail ice mass mixing ratio:

$$M = q_t \Delta_t + \frac{d \cdot C}{V}\left(x - x_{sat}^i\right),$$ (4)

The term $d \cdot C$ in Eq (4) represents the initial contrail volume:

contrail volume (m³) = contrail cross section C (m²) × distance flown d (m)

Contrail cross section is only related to the contrail age and distance flown is aggregated within grid boxes. So when the grid box's horizontal size changes, while the contrail cross-section size remains the same, the distance flown within the grid box would change accordingly.

Assuming the background meteorological conditions remain similar with changes in horizontal resolution, the $\frac{d \cdot C}{V}$ term in Eq (4), the ratio of contrail volume to grid box volume, would not change very much, and therefore the contrail mass mixing ratio and fraction would not be affected much.

This has now been clarified at lines 216-219 as follows:

"We note that using the same cross-sectional area across different spatial resolutions of the two models is expected to have only a negligible effect on young contrail properties. This is because the total contrail volume in a grid box depends not just on the cross-sectional area but also on the grid box aggregated distance flown, which ensures consistency across varying spatial resolutions."

**• L210ff: Could you comment on or estimate the expected differences due to the use of different nudging datasets? Why are two different datasets used in the first place?**

In our simulations we only nudge the wind fields, with UM winds nudged to ERA5 reanalysis and CAM winds nudged to MERRA2 reanalysis. To acknowledge the potential effects of using different datasets for wind nudging, we now add the following in the manuscript at lines 248-250:

"We note that contrail spreading would be affected by the model wind fields, which in our UM and CAM simulations are nudged to ERA5 and MERRA2 reanalysis, respectively. Therefore, differences in the wind fields between these two reanalyses will contribute to variations in the simulated contrail spreading across the two models."

The nudging datasets of the two models are different as they are linked to the standard configurations of the two models, which use ERA5 for the UM and MERRA2 for the CAM simulations as noted in Sect. 2.3. We believe that using these different nudging datasets also allows us to capture the existing uncertainties in contrail cirrus spreading. Providing a comprehensive study of the uncertainty of GCM-simulated contrail cirrus by using the same nudging dataset is beyond the scope of our study.

• **Section 3.1: You mention good agreement between UM and CAM ice supersaturation versus observations, but Figure 1 shows clear differences in the annual zonal mean. Could you comment on this? Which model aligns more closely with observations? Over what time period is the annual zonal mean calculated—one year or more?**

Thank you for pointing this out. Our objective was to highlight the differences in ice supersaturation frequency between the two host climate models, which help explain the significant differences in young contrail fraction and ice water mass. Evaluations of ice supersaturation and humidity in both models have been performed in other previous studies:

"The ice supersaturation generated by the host climate model is key for determining both the microphysical properties and lifetime of the simulated contrail cirrus. Previous evaluation studies show good agreement between simulated UM and CAM ice supersaturation and observations (Chen et al., 2012; Irvine and Shine, 2015). The models' humidity has also been validated against observations and intercompared with other CMIP5 climate models (Jiang et al., 2012)."

We agree it is important to expand a bit the discussion on the differences in ice supersaturation between the two models. We have now updated Fig. 1 in the manuscript to also include a comparison with the ice supersaturation frequency in ERA5 (see updated Fig. 1 below). The ice supersaturation frequency in both models has been calculated on a 1-hour basis to ensure consistency with ERA5 time frequency. We have now also added the following text at lines 265-270:

"Both the UM and CAM capture the general pattern of ice supersaturation found in ERA5 (Figure 1c). However, there are notable overestimations of ice supersaturation across much of the UTLS in both models compared to ERA5 which is known to have a dry bias in the UTLS (Kunz et al., 2014; Wolf et al., 2023). In high-latitude regions below the tropopause, the UM and CAM show ice supersaturation frequencies up to 50% higher than those in ERA5. In the tropical tropopause layer, CAM simulates ice supersaturation frequencies closer to ERA5, while the UM still exhibit higher supersaturation frequencies."

[Figure]

Figure 1. Annual zonal mean frequency of background ice supersaturation simulated by the (a) UM, (b) CAM, and (c) ERA5 for the full year of 2006. The ice supersaturation frequency is calculated on a 1-hour basis. The coloured contour lines represent the annual zonal means of the flight distance (in m/s) in the AEDT air traffic inventory.

The annual zonal mean is calculated for the full year of 2006 – this is now specified in the caption of Fig. 1.

• **L244ff: Why was 2006 chosen for the seasonal cycle, and how representative is that year? Which December (2005 or 2006) was used?**

The choice of the year 2006 for the seasonal cycle was made for consistence with the the air traffic inventory year. We agree that the choice of a particular year will always introduce some uncertainty and we try to account for that by performing perturbed ensembles to account for variations in the background meteorology. We then apply a Student's t-test to indicate statistically significant results under different meteorological conditions, as shown in the figures in Sects. 3.3, 3.4, and 3.5.

December of 2006 was used – all months are in 2006. We now specify this in the caption of Fig. 2 as:

"Panels (d)-(g) show the seasonal mean of the ice supersaturation difference between the UM and CAM for 2006 in December–February (DJF), March–May (MAM), June-August (JJA), and September–November (SON), respectively."

• **Section 3.2: The young contrail is defined as the contrail in the first time step of its life cycle (L258). What differences can be expected if one model has a time step of 20 minutes and the other 30 minutes? Later, it's mentioned that CAM values are normalized by multiplying by 2/3. What if CAM shows no contrails after 30 minutes, but they would still be present after 20 minutes?**

Yes – thank you, this is an important point.

The distance flown d in Eq (4) is calculated as:

Distance flown of the time step = distance flown tendency from the air traffic inventory (m/s) × one model time step length (20min or 30 mins)

Thus, the distance flown calculated from a 30 mins time step will be larger than that from the 20 mins model time step by a factor of 1.5. This is now mentioned in the revised manuscript at lines 335-337:

"As shown in Eq. (4), the young contrail fraction and ice water mass mixing ratio are proportional to the time step length. To ensure comparability of the young contrail quantities, the CAM values were normalised to 30 minutes by a factor of 2/3 to account for the different model time step lengths (i.e. 20 minutes in the UM and 30 minutes in CAM)."

We agree that this normalisation will not account for situations where CAM would show no contrail after 30 minutes, despite potentially being present after 20 minutes. This will contribute to the

difference in simulated young contrail fraction, however will have a negligible effect on the contrail climate impact - a contrail lifetime of 20, 30, or 40 minutes will have virtually no effect on the climate impact of that contrail. To acknowledge this potential contribution to the simulated young contrail fraction, we now add the following in the manuscript at lines 339-340:

"This normalisation may slightly underestimate CAM values, as it does not account for contrails lasting between 20 and 30 minutes period."

• **L283: What do you mean by "high temperature in the Northern Hemisphere"? Are you referring to temperature in the UTLS region?**

Yes, we now clarify this in the manuscript at lines 352-354:

"The minima of contrail cover fraction in boreal summer primarily result from the larger upper tropospheric temperatures in the main regions of intense air traffic (i.e. Northern Hemisphere), which inhibits the formation of contrails."

• **L289: The phrase "This may be due to…" sounds speculative. Can this be substantiated?**

Thank you for pointing this out - we agree it is important to provide more clarity here. We have now reformulated the text in the revised manuscript at lines 357-359 as follows:

"During the boreal summer, the global means of the young contrail cover fraction simulated in the two models are more similar due to the lower UM ice supersaturation over some of the regions with intense air traffic (e.g., Europe and the USA)."

• **L297ff: Differences in CAM results are discussed, showing a factor of 10 when different vertical resolutions are used. How useful is it to compare CAM and UM, which have significantly different vertical resolutions?**

We agree it is important to clarify this. According to the Table 2 from Chen et al. (2012) (also shown below), the global mean young contrail fraction increases with increased vertical resolution in the UTLS. Therefore, the finer vertical resolution (500 m in the UTLS) in the UM likely contributes to a higher contrail fraction compared to CAM (1000m in the UTLS).

| Model level | Vertical resolution in the UTLS | Young contrail fraction global mean |
|---|---|---|
| L30 | ~30hPa/1000m | 6.10E-07 |
| L40 | ~10hPa/300m | 1.73E-06 |
| L54 | ~5hPa/160m | 3.36E-06 |
| L82 | ~2.5hPa/80m | 6.63E-06 |

We now clarify this in the manuscript at lines 375-376:

"In addition, the finer UM vertical resolution in the UTLS also contributes to a larger young contrail cover fraction (Chen et al., 2012)."

• **L308f: What is the temporal and spatial resolution of ECHAM5 in this case?**

The ECHAM5-CCMod at T42L41 resolution has a grid spacing of 2.8° x 2.8° in latitude and longitude, with 41 vertical layers (500 m vertical resolution in the UTLS) (Kurz, 2007), and a time step of 15 minutes. These are different to those of the UM and CAM and they will have a slight contribution to differences in simulated contrail fractions. However, the main contributor to this difference is the

overwhelmingly different definition of young contrail fraction used in ECHAM, compared to UM or CAM. As noted in Sect. 3.2, the contrail fraction in ECHAM corresponds to contrail cirrus with an optical depth threshold of at least 0.05, which also includes contrails older than one model time step.

**• Section 3.4: The description of the scaling factor is brief. Could you explain how you arrived at the value of 4900? Is this value representative for regions with less traffic?**

To obtain the most suitable scaling factor, we used a trial-and-error method based on several simulations spanning scaling factors between 1000 and 20000. The 4900 scaling factor was derived based on the simulated optical depth over the European region, where the contrail signal is relatively stronger (compared to other regions) due to the higher air traffic. For regions with less traffic, where the contrail signal is much weaker, it is very difficult to obtain statistically significant results, even when employing a large number of ensemble simulations. This does indeed mean that this scaling factor is not necessarily representative for regions with less traffic.

To acknowledge this, the revised version of the manuscript states at lines 437-441:

"To obtain the most suitable scaling factor for each optical depth reference value, we used a trial-and-error method based on several simulations spanning scaling factors between 1000 and 20000. We used the European region (35°N – 60°N latitude and 10°W – 25°E longitude) as benchmark due to its large air traffic and therefore larger statistical significance. As a result, the scaling factors may not be representative for areas with lower air traffic density, where obtaining statistically significant results is more challenging."

**• Figure 6 Caption: What are the "annual mean simulated contrail driven changes" compared to?**

Thank you for pointing to this source of confusion. The 'annual mean simulated contrail-driven changes' refer to the average annual changes caused by contrails, calculated as the difference between simulations with contrails and those without contrails. We now clarify this in the caption of Fig. 6 as follows:

"These values are calculated as the difference between simulations with contrails and those without contrails."

**• L393: The phrase "is likely due to…" sounds speculative. Can this be substantiated?**

We agree and we have now modified the text in the revised manuscript at lines 473-475 as follows:

"In addition to the scaling of young contrails mass in the UM, the factor of 2 difference between these values is also due to the different radiative transfer schemes and different cloud microphysical process rates in the UM and CAM."

**• Summary (L416): You mention the use of the same contrail scheme in two different host climate models. However, disentangling the differences due to one- and two-moment microphysics is challenging enough. Including different climate models with varying resolutions seems to skip a necessary step (as mentioned above).**

We believe that by using the same contrail scheme (i.e. a scheme where contrails are diagnosed within a single model timestep and then passed to the model cloud scheme) in the standard configurations of two widely used climate models is an important contribution. The fact that these standard configurations are different (e.g. spatial and temporal resolutions, nudging) and their consequences are acknowledged in our discussion and is in itself an important consideration of how future assessments of contrail cirrus climate impacts need to be interpreted. We believe that our paper

highlights several key differences in the contrail simulations between the host models, including young contrail estimates, cloud fraction changes, and contrail cirrus ERF. The very small contrail cirrus ERF in the UM, despite its larger young contrail fraction and ice mass compared to CAM, is primarily due to differences in cloud microphysics, specifically the treatment of contrail ice number concentration in the two host models. In the UM, contrails are added to natural clouds with microphysical properties similar to those of natural clouds. Given that young contrail ice particles are much smaller than natural cloud particles, this approach is not ideal. As a result, contrails in the UM form with a lower number concentration and larger particle size than in CAM, which likely leads to increased sedimentation rates and shorter lifetimes. This accounts for the negligible contrail cirrus ERF in the UM. Additionally, variations in background meteorology and resolution affect young contrail estimates, while opposing cloud fraction changes are attributed to differences in cloud microphysics between the models.

This has now been clarified in the manuscript at lines 524-526 as follows:

"Another source of uncertainty arises from the differences in configurations (e.g. spatial and temporal resolutions, nudging) between the UM and CAM. In this study, both these widely used climate models are employed in their standard configurations, which are also likely to be used in future contrail studies."

**• L429ff and L373ff: You write that the contrail cirrus is misrepresented in UM for understandable reasons, but it should be shown more clearly that CAM provides a more realistic representation, especially since UM's optical depth is matched to CAM's values**

We do not think that we can make this point. Here we cannot provide an entirely independent UM estimate for contrail cirrus ERF due to the current shorcomings of the UM cloud scheme. However, by reporting the UM estimates for contrail cirrus ERF when matching different simulated optical depths, we provide an additional insight into how model characteristics contribute to contrail cirrus ERF uncertainty. We compare the UM simulated contrail cirrus optical depth with both CAM and ECHAM (as the other GCMs that simulate contrails) to better reveal this existing uncertainty range. The differences in optical depth across these models stem from both the physics and contrail parameterisations of host models. Both UM and CAM implement the same contrail parameterization (Chen et al., 2012), whereas ECHAM follows a different approach (Burkhardt and Kärcher, 2009). There is no clear benchmark contrail cirrus ERF to definitively determine which model provides the most accurate contrail ERF estimates, with large uncertainties remaining, as noted in Lee et al. (2021).

**• L443 (Future Work): It appears that microphysics is recognized as the greatest uncertainty, and improving UM with a new two-moment microphysics scheme is suggested. If microphysics is indeed the primary factor driving differences, the title of the manuscript should reflect this focus, perhaps as "Impact of Microphysics on Contrail Cirrus Radiative Forcing."**

As we mentioned in our general comment response, we believe that our paper highlights several important differences in contrail simulations between two host climate models – which in itself is something that has not been done before. While the microphysics scheme is indeed a very important one, there are other important findings that are presented and discussed in this study, such as the differences in young contrail estimates, the model cloud fraction response due to contrails, and contrail cirrus ERF. The small contrail cirrus ERF in the UM, driven by differences in cloud microphysics, represents only one aspect of the study. We therefore believe that the current title is an accurate representation of our paper and its scientific contribution.

**Typos, Format, etc.:**

Thank you for the detailed comments regarding the typos, formatting, and other corrections. These have been addressed in the revised manuscript.

**L150ff: In LaTeX math mode, use \textrm for text and \unit{} for units. Ensure consistent typesetting and spacing between different units.**

This has now been addressed in the revised version. Just to clarify that the "spacings" before the commas in Line 167 are due to the document's justification format, and there are no actual spacings before the commas.

**References**

Bock, L. and Burkhardt, U.: Reassessing properties and radiative forcing of contrail cirrus using a climate model, J. Geophys. Res. Atmos., 121, 9717-9736, https://doi.org/10.1002/2016JD025112, 2016.

Brown, F., Folberth, G., Sitch, S., Artaxo, P., Bauters, M., Boeckx, P., Cheesman, A. W., Detto, M., Komala, N., Rizzo, L., Rojas, N., dos Santos Vieira, I., Turnock, S., Verbeeck, H., and Zambrano, A.: Performance evaluation of UKESM1 for surface ozone across the pan-tropics, EGUsphere, 2024, 1-26, 10.5194/egusphere-2023-2937, 2024.

Brown-Steiner, B., Hess, P. G., and Lin, M. Y.: On the capabilities and limitations of GCCM simulations of summertime regional air quality: A diagnostic analysis of ozone and temperature simulations in the US using CESM CAM-Chem, Atmospheric Environment, 101, 134-148, https://doi.org/10.1016/j.atmosenv.2014.11.001, 2015.

Burkhardt, U. and Kärcher, B.: Process-based simulation of contrail cirrus in a global climate model, J. Geophys. Res. Atmos., 114, https://doi.org/10.1029/2008JD011491, 2009.

Chen, C. C. and Gettelman, A.: Simulated radiative forcing from contrails and contrail cirrus, Atmos. Chem. Phys., 13, 12525-12536, 10.5194/acp-13-12525-2013, 2013.

Chen, C. C., Gettelman, A., Craig, C., Minnis, P., and Duda, D. P.: Global contrail coverage simulated by CAM5 with the inventory of 2006 global aircraft emissions, J. Adv. Model. Earth Syst., 4, https://doi.org/10.1029/2011MS000105, 2012.

Danabasoglu, G., Lamarque, J.-F., Bacmeister, J., Bailey, D. A., DuVivier, A. K., Edwards, J., Emmons, L. K., Fasullo, J., Garcia, R., Gettelman, A., Hannay, C., Holland, M. M., Large, W. G., Lauritzen, P. H., Lawrence, D. M., Lenaerts, J. T. M., Lindsay, K., Lipscomb, W. H., Mills, M. J., Neale, R., Oleson, K. W., Otto-Bliesner, B., Phillips, A. S., Sacks, W., Tilmes, S., van Kampenhout, L., Vertenstein, M., Bertini, A., Dennis, J., Deser, C., Fischer, C., Fox-Kemper, B., Kay, J. E., Kinnison, D., Kushner, P. J., Larson, V. E., Long, M. C., Mickelson, S., Moore, J. K., Nienhouse, E., Polvani, L., Rasch, P. J., and Strand, W. G.: The

Community Earth System Model Version 2 (CESM2), J. Adv. Model. Earth Syst., 12, e2019MS001916, https://doi.org/10.1029/2019MS001916, 2020.

Henry, M., Haywood, J., Jones, A., Dalvi, M., Wells, A., Visioni, D., Bednarz, E. M., MacMartin, D. G., Lee, W., and Tye, M. R.: Comparison of UKESM1 and CESM2 simulations using the same multi-target stratospheric aerosol injection strategy, Atmos. Chem. Phys., 23, 13369-13385, 10.5194/acp-23-13369-2023, 2023.

Kurz, C.: Entwicklung und Anwendung eines gekoppelten Klima-Chemie-Modellsystems: Globale Spurengastransporte und chemische Umwandlungsprozesse, München, Univ., Diss., 2006, 2007.

Lee, D. S., Fahey, D. W., Skowron, A., Allen, M. R., Burkhardt, U., Chen, Q., Doherty, S. J., Freeman, S., Forster, P. M., Fuglestvedt, J., Gettelman, A., De León, R. R., Lim, L. L., Lund, M. T., Millar, R. J., Owen, B., Penner, J. E., Pitari, G., Prather, M. J., Sausen, R., and Wilcox, L. J.: The contribution of global aviation to anthropogenic climate forcing for 2000 to 2018, Atmospheric Environment, 244, 117834, https://doi.org/10.1016/j.atmosenv.2020.117834, 2021.

Ratcliffe, N. G., Ryder, C. L., Bellouin, N., Woodward, S., Jones, A., Johnson, B., Weinzierl, B., Wieland, L. M., and Gasteiger, J.: Long range transport of coarse mineral dust: an evaluation of the Met Office Unified Model against aircraft observations, EGUsphere, 2024, 1-32, 10.5194/egusphere-2024-806, 2024.

Sellar, A. A., Jones, C. G., Mulcahy, J. P., Tang, Y., Yool, A., Wiltshire, A., O'Connor, F. M., Stringer, M., Hill, R., Palmieri, J., Woodward, S., de Mora, L., Kuhlbrodt, T., Rumbold, S. T., Kelley, D. I., Ellis, R., Johnson, C. E., Walton, J., Abraham, N. L., Andrews, M. B., Andrews, T., Archibald, A. T., Berthou, S., Burke, E., Blockley, E., Carslaw, K., Dalvi, M., Edwards, J., Folberth, G. A., Gedney, N., Griffiths, P. T., Harper, A. B., Hendry, M. A., Hewitt, A. J., Johnson, B., Jones, A., Jones, C. D., Keeble, J., Liddicoat, S., Morgenstern, O., Parker, R. J., Predoi, V., Robertson, E., Siahaan, A., Smith, R. S., Swaminathan, R., Woodhouse, M. T., Zeng, G., and Zerroukat, M.: UKESM1: Description and Evaluation of the U.K. Earth System Model, 11, 4513-4558, https://doi.org/10.1029/2019MS001739, 2019.

Son, S.-W., Han, B.-R., Garfinkel, C. I., Kim, S.-Y., Park, R., Abraham, N. L., Akiyoshi, H., Archibald, A. T., Butchart, N., Chipperfield, M. P., Dameris, M., Deushi, M., Dhomse, S. S., Hardiman, S. C., Jöckel, P., Kinnison, D., Michou, M., Morgenstern, O., O'Connor, F. M., Oman, L. D., Plummer, D. A., Pozzer, A., Revell, L. E., Rozanov, E., Stenke, A., Stone, K., Tilmes, S., Yamashita, Y., and Zeng, G.: Tropospheric jet response to Antarctic ozone depletion: An update with Chemistry-Climate Model Initiative (CCMI) models, Environ. Res. Lett., 13, 054024, 10.1088/1748-9326/aabf21, 2018.

---

## Referee Report (RR1)

**Comments on "Impact of host climate model on contrail cirrus effective radiative forcing estimates" by Zhang et al. (2024), revised version**

I greatly appreciate the effort that was clearly put into improving the manuscript. However, I was surprised by the fact that the description of the new figure 3 made it into the revised manuscript two times in slightly different versions. I therefore wonder whether the authors read their manuscript before handing it in.

Some further comments:

- Line 41: Why was "ICAO" converted to lower case here? If this was on purpose, it is still upper case in line 37.
- I found "parameterisation" in four different spellings on page 4.
- Line 133ff: The abbreviation "RH" is only used in these few lines and has not been introduced.
- Line 176: "ambient relative humidity" -> "ambient relative humidity with respect to liquid water" Just for clarity, since the next sentence is on supersaturation with respect to ice.
- Line 185: "aviation water vapour emission" -> "ambient water vapour"
- Line 473: "In addition to the scaling of young contrails mass in the UM, the factor of 2 difference between these values is also due to..." -> "The factor of 2 difference between these values may partly be caused by the fact that our scaling approach only affected the contrails in the first timestep of their lifecycle. Besides that, it might also be due to..."
  I think that the difference cannot be attributed to differences in the microphysics and radiation scheme alone, as the contrail ice water path has only been aligned for young contrails. I therefore would also question the "factor of 2 uncertainty in contrail cirrus ERF due to differences in the model microphysics and radiation schemes" in line 521f.